# GLYPH-SR: Can We Achieve Both High-Quality Image Super-Resolution and High-Fidelity Text Recovery via VLM-Guided Latent Diffusion Model?

## Abstract

Image super-resolution (SR) is fundamental to many vision systems—from surveillance and autonomy to document analysis and retail analytics—because recovering high-frequency details, especially scene-text, enables reliable downstream perception. scene-text, i.e., text embedded in natural images such as signs, product labels, and storefronts, often carries the most actionable information; when characters are blurred or hallucinated, optical character recognition (OCR) and subsequent decisions fail even if the rest of the image appears sharp. Yet previous SR research has often been tuned to distortion (PSNR/SSIM) or learned perceptual metrics (LPIPS, MANIQA, CLIP-IQA, MUSIQ) that are largely insensitive to character-level errors. Furthermore, studies that do address text SR often focus on simplified benchmarks with isolated characters, overlooking the challenges of text within complex natural scenes. As a result, scene-text is effectively treated as generic texture. For SR to be effective in practical deployments, it is therefore essential to explicitly optimize for both text legibility and perceptual quality. We present GLYPH-SR, a vision–language-guided diffusion framework that aims to achieve both objectives jointly. GLYPH-SR utilizes a Text-SR Fusion ControlNet (TS-ControlNet) guided by OCR data, and a ping-pong scheduler that alternates between text- and scene-centric guidance. To enable targeted text restoration, we train these components on a synthetic corpus while keeping the main SR branch frozen. Across SVT, SCUT-CTW1500, and CUTE80 at ×4 and ×8, GLYPH-SR improves OCR $F_1$ by up to +15.18 percentage points over diffusion/GAN baselines (SVT ×8, OpenOCR) while maintaining competitive MANIQA, CLIP-IQA, and MUSIQ. GLYPH-SR is designed to satisfy both objectives simultaneously—high readability and high visual realism—delivering SR that looks right and reads right. We provide code, pretrained models, the synthetic corpus with generation scripts, and an evaluation suite to support reproducibility.

## 1 Introduction

Image super-resolution (SR),[1] which reconstructs high-resolution (HR) images from low-resolution (LR) inputs, is critical for applications like autonomous driving where clear details are paramount. While conventional SR aims to improve perceptual quality, we argue that for many real-world scenarios, ensuring the text legibility of scene-text (e.g., on signs, license plates) is equally, if not more, important. Accurately restoring characters is crucial, as failures in legibility can compromise downstream tasks like optical character recognition (OCR), regardless of the overall image sharpness.

### 1.1 An Overlooked Challenge in Image SR: Achieving High Scene-Text Fidelity

However, achieving this level of text fidelity remains an overlooked challenge in most conventional SR frameworks. Two systemic biases explain why text often degrades in existing SR models (e.g.,

---

[1]Throughout this paper, we will use *image SR* and *SR* interchangably whenever there is no ambiguity.

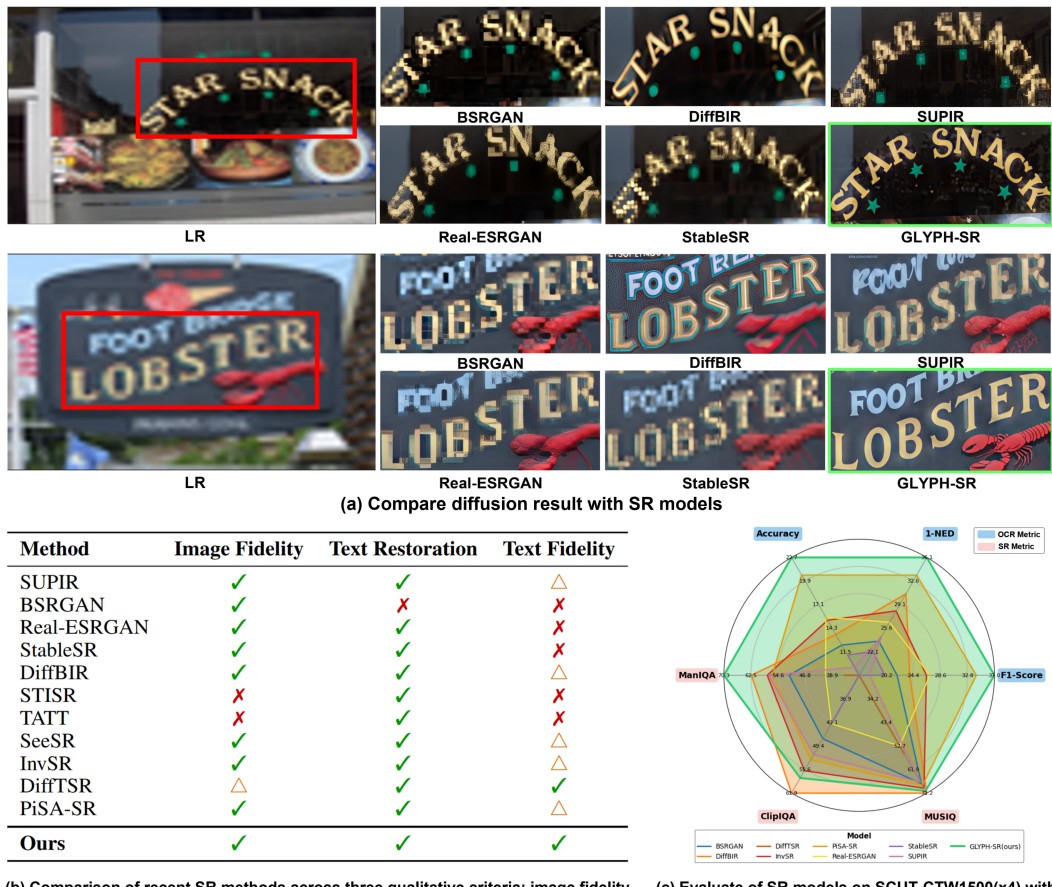

Figure 1: Qualitative and quantitative comparisons of our GLYPH-SR with other competing SR methods, demonstrating superior text fidelity and OCR $F_1$ score.

StableSR Wang et al. (2024), DiffBIR Lin et al. (2024), InvSR Yue et al. (2025)) despite strong perceptual scores:

(a) **Metric Bias.** Standard full-reference distortion metrics (PSNR/SSIM) and learned/no-reference perceptual metrics (LPIPS, MANIQA, CLIP-IQA, MUSIQ) aggregate quality globally and are dominated by area; small text regions (often well below $1\%$ of the image) therefore contribute little, so character corruption is weakly penalized.

(b) **Objective Bias.** Common training losses prioritize appearance similarity and treat characters as generic high-frequency texture rather than discrete semantic units required by OCR.

In practice these biases surface as two failure modes (Fig. 1 (a)): *(i) Hallucination*—methods optimized for perceptual realism may produce sharp but incorrect characters, harming OCR; *(ii) Conservative restoration*—others preserve the blurry input to avoid artifacts, yielding limited SR gains alongside mediocre perceptual quality. As a result, few approaches simultaneously enhance visual realism and ensure text legibility—an essential requirement for OCR-dependent applications.

## 1.2 CONTRIBUTIONS

We address scene-text SR as a *bi-objective* problem—optimizing both **visual quality** and **text legibility**—and present **GLYPH-SR**, a vision–language guided diffusion framework that achieves both. Our key technical contributions and breakthroughs in this work include the followings:

- **Bi-Objective Formulation & Dual-Axis Evaluation.** We explicitly cast SR in text-rich scenes as the joint optimization of *image quality* and *readability*, and standardize a *dual-axis* protocol that reports perceptual SR metrics (MANIQA, CLIP-IQA, MUSIQ) *together with* OCR-aware measures (word/character accuracy, edit distance, $F_1$), ensuring that small text regions are not underweighted.

- **Text-SR Fusion ControlNet with Time-Balanced Guidance.** We introduce a dual-branch **TS-ControlNet** that fuses *token-level OCR strings with verbalized locations* $S_{\text{TXT}}$ and a *scene caption* $S_{\text{IMG}}$. The SR branch is frozen while the text branch is fine-tuned; residual mixing injects complementary cues into the LDM without disrupting its generative prior. A lightweight **ping–pong** scheduler $\lambda_t$ alternates text-centric and image-centric conditioning along the denoising trajectory, and coherently modulates both *embedding fusion* and *residual injection*.

- **Factorized Synthetic Corpus & Comprehensive Validation.** We build a four-partition synthetic corpus that independently perturbs glyph quality and global image quality, enabling targeted text restoration while keeping the SR branch frozen. Across SVT, SCUT-CTW1500, and CUTE80 at $\times 4/\times 8$, **GLYPH-SR** improves OCR $F_1$ by up to **+15.18 pp** over strong diffusion/GAN baselines while maintaining competitive MANIQA, CLIP-IQA, and MUSIQ. We release code, pretrained models, data-generation scripts, and an evaluation suite to support reproducibility.

## 2 RELATED WORKS

**SR via Deep Learning.** Early CNN methods such as SRCNN Dong et al. (2015), EDSR Lim et al. (2017), and RCAN Zhang et al. (2018b), and later transformer models like SwinIR Liang et al. (2021), substantially advanced distortion-oriented SR; yet they primarily optimize pixel fidelity rather than semantic fidelity in small, text-bearing regions. Adversarially trained SR has improved perceptual realism on in-the-wild images; representative examples include BSRGAN Zhang et al. (2021) and Real-ESRGAN Wang et al. (2021).

**General-Purpose SR Models.** Diffusion-based SR has recently shown strong stability and realism. Foundational approaches such as DiffBIR Lin et al. (2024), ConsisSR Gu et al. (2024) and StableSR Wang et al. (2024) couple LR conditioning with powerful diffusion priors, and subsequent work incorporates richer priors or auxiliary conditions. Some methods exploit text-based prompts: SeeSR Wu et al. (2024) uses semantic prompts, while PromptSR Chen et al. (2023) directly injects text prompts to improve performance. SUPIR Yu et al. (2024b) also leverages text prompts, but combines them with large-scale pretrained backbones and restoration-guided sampling. Other approaches include InvSR Yue et al. (2025), which enables flexible guidance/sampling, and PISA-SR Sun et al. (2025), which further advances controllability. As illustrated in Fig. 1(b), explicit character-level integrity is seldom a primary optimization target in general-purpose diffusion SR. Consequently, as further substantiated by the quantitative benchmarks in Fig. 1(c), there is a notable scarcity of methods that holistically address both general image fidelity and text-specific restoration metrics.

**Text-Focused SR.** Text-centric SR aims to enhance readability with text-aware priors or recognition-aware objectives. Representative methods include TATT Ma et al. (2022), STISR Noguchi et al. (2024), MARCONet Li et al. (2023; 2025) and Stroke-Aware SR Chen et al. (2022). While effective on word/line crops, these approaches often assume simplified settings and can underperform on full natural scenes where text must be preserved together with surrounding content. While recent works such as Min et al. (2025) address text within scenes, they do not sufficiently address scenarios involving severe image quality degradation.

Thus, developing a scene text SR method that can simultaneously enhance overall image fidelity and ensure precise text restoration under severe low-resolution conditions remains a significant and open challenge.

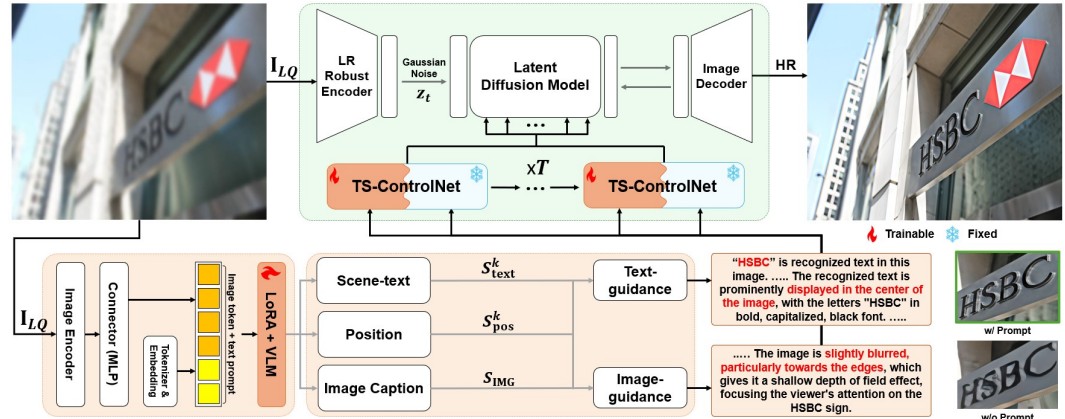

Figure 2: Overview of the proposed GLYPH-SR architecture.

# 3 OUR APPROACH: GLYPH-SR

## 3.1 MODEL ARCHITECTURE

**Overview.** Fig. 2 depicts the proposed **GLYPH-SR** pipeline. Given an LR image $\mathbf{I}_{\text{LR}} \in \mathbb{R}^{H \times W \times C}$, an LR-robust conditioner of a pretrained latent diffusion model (LDM) Rombach et al. (2022) extracts multi-scale features $f_{\text{LR}}$ used for conditioning. Our **Text–SR Fusion ControlNet (TS-ControlNet)** then injects complementary restoration cues while preserving the generative prior of the LDM. Finally, an Elucidated Diffusion Model (EDM) sampler Karras et al. (2022) drives the reverse process in latent space toward a high-resolution reconstruction. However, when guidance is provided only in a holistic form, small text regions may still be treated as generic high-frequency textures rather than semantically meaningful glyphs, which can yield imperfect character restoration.

**Condition Decomposition.** To address this limitation, we explicitly separate the guidance into (i) **image-oriented** and (ii) **text-oriented** signals.

- **Image-Oriented Guidance.** A scene-level caption $\mathcal{S}_{\text{IMG}}$ summarizes global attributes such as illumination, composition, and depth-of-field, and is used to encourage holistic perceptual quality.

- **Text-Oriented Guidance.** A dedicated OCR module detects $K$ text instances and returns position–text pairs $\{(\mathcal{S}_{\text{text}}^k, \mathcal{S}_{\text{pos}}^k)\}_{k=1}^K$. Each pair is converted into a structured natural-language prompt, e.g. "HSBC is displayed at the center of the image," and passed to the text branch.

As shown in Fig. 3(b), simply separating $\mathcal{S}_{\text{IMG}}$ and $\{(\mathcal{S}_{\text{text}}^k, \mathcal{S}_{\text{pos}}^k)\}_{k=1}^K$ improves text fidelity but can degrade non-text regions, motivating our subsequent guidance-fusion strategy and the ping–pong scheduler that alternates text-centric and scene-centric guidance.

**Text–SR Fusion ControlNet.** To balance the two objectives—image quality and text legibility—we introduce the *Text-SR Fusion ControlNet* (TS-ControlNet), which merges glyph-level semantic priors with global SR guidance (Fig. 3c). During training, the LDM backbone and the SR branch of TS-ControlNet are frozen, and only the text branch is updated, improving text legibility while preserving overall image quality.

Given image data $\mathbf{I}$, we obtain the clean target latent $z_0 = \text{enc}(\mathbf{I})$ via the VAE encoder. We then sample a timestep $t \sim \mathcal{U}\{1, \dots, T\}$ and noise $\varepsilon \sim \mathcal{N}(0, \mathbf{I})$, and construct the noised latent by the standard DDPM forward process Ho et al. (2020):

$$z_t = \sqrt{\bar{\alpha}_t}\, z_0 + \sqrt{1 - \bar{\alpha}_t}\, \varepsilon, \qquad \bar{\alpha}_t = \prod_{s=1}^t (1 - \beta_s).$$

The diffusion model $\mathcal{D}_\theta$ predicts the noise residual conditioned on two control streams: (i) $\mathcal{C}_{\text{SR}}$, a spatial condition from a frozen SR-ControlNet that guides the overall structure based on the low-

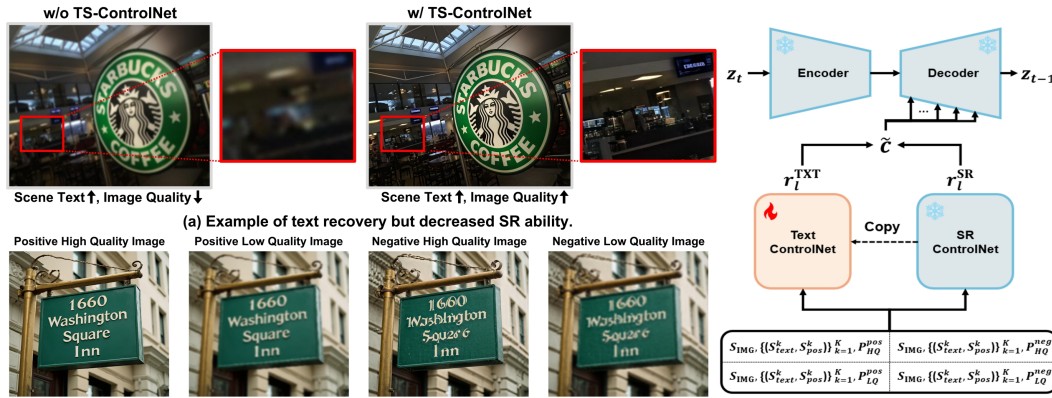

Figure 3: Text-centric fine-tuning framework: (a) trade-off between scene-text fidelity and overall image quality according to guidance; (b) four synthetic training subsets with matched prompts; (c) TS-ControlNet architecture.

resolution input image $\mathcal{S}_{\text{IMG}}$, and (ii) $\mathcal{C}_{\text{TXT}}$, a textual condition from a trainable Text-ControlNet that controls the rendering of text based on a set of OCR-derived text-position pairs $\mathcal{S}_{\text{TXT}}$.

At inference, we start from $z_T$ and use the EDM sampler Karras et al. (2022) with the same conditions to obtain the HR latent, which is then decoded to the image domain.

**Diffusion Loss with Residual Injection.** The frozen SR-ControlNet and the trainable Text-ControlNet produce residual hierarchies. We blend them before injection via

$$c = \frac{1}{2} s_{\text{CTRL}} \Big[ \mathcal{C}_{\text{SR}}\big(z_t; \phi_{\text{img}}(\mathcal{S}_{\text{IMG}} + P)\big) + \mathcal{C}_{\text{TXT}}\big(z_t; \phi_{\text{txt}}(\mathcal{S}_{\text{TXT}} + P)\big) \Big]. \tag{1}$$

where $s_{\text{CTRL}}$ is a global scaling factor and $P$ denotes the restoration guide prompt.

The diffusion backbone $\mathcal{D}_\theta$ then predicts the residual noise, and we optimize TS-ControlNet with the standard $\varepsilon$-prediction objective:

$$\mathcal{L}_{\text{text}} = \mathbb{E}_{z_0, t, \varepsilon} \big\| \varepsilon - \mathcal{D}_\theta(z_t, t, c) \big\|_2^2. \tag{2}$$

**Synthetic Fine-Tuning Dataset.** To disentangle text legibility from holistic perceptual quality, we synthesize four mutually exclusive subsets $\{\mathbf{I}_{\text{HQ}}^{\text{pos}}, \mathbf{I}_{\text{LQ}}^{\text{pos}}, \mathbf{I}_{\text{HQ}}^{\text{neg}}, \mathbf{I}_{\text{LQ}}^{\text{neg}}\}$. All synthetic data are generated from the same raw text, but for training purposes, the image quality is intentionally reduced or only the text within the images is distorted. As shown in Fig. 3 (b). To train *TS-ControlNet*, we defined the following guide prompt.

- **Positive–Text / High-Quality** ($P_{\text{HQ}}^{\text{pos}}$). Perfect image quality with perfectly preserved character outlines and precise positioning.
- **Negative–Text / High-Quality** ($P_{\text{HQ}}^{\text{neg}}$). Intentionally damaged character outlines and precise positioning, but good image quality.
- **Positive–Text / Low-Quality** ($P_{\text{LQ}}^{\text{pos}}$). Poor image quality, but preserved character outlines and precise positioning.
- **Negative–Text / Low-Quality** ($P_{\text{LQ}}^{\text{neg}}$). Image quality is poor and character outlines and exact positions are intentionally damaged.

Each sample is encoded into a *composite conditioning tuple* for the *TS-ControlNet*:

$$\underbrace{z_\star^\diamond}_{\text{image latent}} \oplus \underbrace{\psi(\mathcal{S}_{\text{IMG}})}_{\text{scene caption}} \oplus \underbrace{\psi(\{(\mathcal{S}_{\text{text}}^k, \mathcal{S}_{\text{pos}}^k)\}_{k=1}^K)}_{\text{text cues}} \oplus \underbrace{P_\star^\diamond}_{\text{guide prompt}}, \qquad \diamond \in \{\text{pos}, \text{neg}\}, \ \star \in \{\text{HQ}, \text{LQ}\}.$$

Here, $z_\star^\diamond = \text{Enc}(\mathbf{I}_\star^\diamond)$ is the first-stage latent of the synthetic image $\mathbf{I}_\star^\diamond$, and $\psi(\cdot)$ denotes the frozen CLIP text encoder. Note that, to explicitly inform the model when incorrect text has been generated,

the text-position pairs $\{(\mathcal{S}_{\text{text}}^k, \mathcal{S}_{\text{pos}}^k)\}_{k=1}^K$ are always extracted from the positive-text, high-quality image dataset.

## 3.2 Text–Image Balancing Scheduler

Although the dedicated *TS-ControlNet* injects glyph-centric features, the temporal allocation between text and image guidance along the diffusion trajectory is critical. We therefore introduce a scheduler $\mathcal{T}_{\text{sched}} : \{0, \ldots, T\} \to [0, 1]$ that dynamically reweights the two guidance streams via a time-dependent coefficient $\lambda_t$.

**Step update with mixed guidance.** Let $z_t$ be the latent at diffusion step $t$ (sampling proceeds from $t = T$ down to 0). Given a mixed embedding $e^t$ (Eq. 4), we form a classifier-free guided noise estimate (Eq. 5) and then update

$$z_{t-1} = z_t - \eta_t \widehat{\epsilon}_t, \tag{3}$$

where $\eta_t$ is a step size (a function of the noise level $\sigma_t$ in our EDM-based solver). At inference we initialize $z_T \sim \mathcal{N}(0, \sigma_T^2 \mathbf{I})$ and apply the EDM sampler Karras et al. (2022) with the same conditions over $T$ steps.

We encode scene-level and text-level prompts separately and fuse them as

$$e_{\text{img}} = W_{\text{img}} \phi_{\text{img}}(\mathcal{S}_{\text{IMG}}), \quad e_{\text{txt}} = W_{\text{txt}} \phi_{\text{txt}}(\{(\mathcal{S}_{\text{text}}^k, \mathcal{S}_{\text{pos}}^k)\}_{k=1}^K), \quad e^t = (1-\lambda_t)\, e_{\text{txt}} + \lambda_t\, e_{\text{img}}, \tag{4}$$

where $\phi_{\text{img}}$ and $\phi_{\text{txt}}$ are text encoders (kept frozen), and $W_{\text{img}}, W_{\text{txt}}$ are linear projections to a shared embedding space. The guided residual is computed via classifier-free guidance:

$$\widehat{\epsilon}_t = (1 + \omega)\, \mathcal{D}_\theta(z_t, t, e^t) - \omega\, \mathcal{D}_\theta(z_t, t, \varnothing), \tag{5}$$

with guidance scale $\omega$. Consistently, the same $\lambda_t$ also modulates residual injection (cf. Eq. 1) as a time-varying blend $\tilde{r}_l(t) = s_{\text{CTRL}}\big[(1 - \lambda_t)\, r_l^{\text{TXT}} + \lambda_t\, r_l^{\text{SR}}\big]$.

**Binary Ping-Pong Policy.** We found that a *binary* schedule that alternates between text-centric ($\lambda_t = 0$) and image-centric ($\lambda_t = 1$) guidance is effective:

$$\lambda_t = \begin{cases} 0, & \text{if } \left\lfloor \frac{t - t_0}{\tau} \right\rfloor \bmod 2 = 0, \\ 1, & \text{otherwise,} \end{cases} \tag{6}$$

where $\tau \in \mathbb{N}$ is the toggle period (default $\tau = 1$) and $t_0$ is an optional offset. Intuitively, the text-focused phases inject precise glyph cues, while the image-focused phases stabilize global structure and appearance. We also experimented with continuous ramps $\lambda_t = g(\sigma_t)$ (e.g., noise-level monotone schedules), but the square-wave "ping–pong" yielded the best OCR $F_1$ at similar perceptual quality (see Appendix C).

## 4 Experiments

### 4.1 Experimental Setup

We evaluate our method along two axes: semantic text restoration and perceptual SR quality. We report OCR-based $\mathbf{F_1}$ scores Chng et al. (2019) to quantify semantic correctness. Pixel-wise fidelity is measured by MANIQA Yang et al. (2022), CLIP-IQA Wang et al. (2023), and MUSIQ Ke et al. (2021) (see Sec. A.1). Experiments are conducted on three representative scene-text benchmarks (details in Sec. B.2.1): SCUT-CTW1500 Liu et al. (2019), CUTE80 Risnumawan et al. (2014), and SVT Wang et al. (2011). We adopt *Juggernaut-XL* as the LDM backbone and fine-tune it on our synthetic corpus generated with LLaVA-NeXT Liu et al. (2024), Nunchaku Cruanes et al. (2016), and SUPIR Yu et al. (2024b). Full data-generation pipelines and hyper-parameters and setup are detailed in Appendix B.

### 4.2 Evaluation Results

As presented in Table 1, prior methods typically sacrifice one objective for the other due to the trade-off between text fidelity and perceptual quality. For instance, **DiffBIR** achieves high perceptual scores

Table 1: Quantitative comparison of OCR $F_1$-scores and SR quality metrics across datasets and models. Red and blue indicate the best and second-best scores, respectively. Real-ESRGAN has been included to match the full benchmark results.

| Dataset | Model | OpenOCR | | GOT-OCR | | LLaVA-NeXT | | MANIQA | | CLIP-IQA | | MUSIQ | |
|---|---|---|---|---|---|---|---|---|---|---|---|---|---|
| | | ×4 | ×8 | ×4 | ×8 | ×4 | ×8 | ×4 | ×8 | ×4 | ×8 | ×4 | ×8 |
| SVT | LR | 48.65 | 8.49 | 66.89 | 27.78 | 70.08 | 42.79 | 20.45 | 19.81 | 17.06 | 44.07 | 26.31 | 22.96 |
| | BSRGAN | 53.96 | 14.61 | 58.66 | 13.12 | 68.50 | 25.56 | 38.16 | 37.14 | 39.63 | 37.58 | 66.25 | 62.83 |
| | DiffBIR | 38.73 | 16.70 | 42.33 | 18.55 | 45.19 | 22.32 | 47.82 | 45.54 | 58.66 | 53.20 | 71.18 | 64.11 |
| | DiffTSR | 19.35 | 10.28 | 22.51 | 10.72 | 29.23 | 15.87 | 21.34 | 21.39 | 27.69 | 26.39 | 46.24 | 43.96 |
| | InvSR | 57.79 | 17.12 | 60.96 | 21.15 | 65.00 | 21.54 | 46.78 | 32.51 | 57.30 | 50.83 | 70.81 | 51.69 |
| | MARCONet | 0.00 | 0.00 | 0.25 | 0.00 | 0.00 | 0.00 | 30.92 | 30.84 | 24.43 | 24.76 | 27.26 | 27.02 |
| | MARCONet++ | 50.05 | 10.28 | 59.88 | 13.97 | 65.90 | 22.32 | 29.31 | 20.97 | 19.82 | 8.44 | 49.20 | 38.06 |
| | PiSA-SR | 63.30 | 17.53 | 65.23 | 24.05 | 67.75 | 37.76 | 37.41 | 34.02 | 44.30 | 18.39 | 61.87 | 30.24 |
| | Real-ESRGAN | 59.15 | 17.73 | 67.32 | 23.29 | 72.53 | 30.83 | 31.16 | 28.38 | 28.58 | 17.86 | 51.14 | 43.01 |
| | SwinIR | 54.61 | 14.61 | 63.53 | 20.75 | 73.03 | 30.48 | 26.32 | 22.05 | 44.50 | 26.68 | 34.55 | 30.33 |
| | StableSR | 59.88 | 20.95 | 63.76 | 24.43 | 73.91 | 43.24 | 24.75 | 23.16 | 32.18 | 23.38 | 24.44 | 16.22 |
| | SUPIR | 58.41 | 33.61 | 61.90 | 35.96 | 62.14 | 36.78 | 42.36 | 40.17 | 48.42 | 45.06 | 67.55 | 65.20 |
| | TAIR | 27.23 | 21.54 | 30.13 | 23.48 | 32.58 | 26.68 | 31.99 | 31.99 | 29.12 | 29.49 | 54.34 | 54.27 |
| | **GLYPH-SR** | **67.54** | **48.79** | **71.72** | **56.16** | **73.22** | **58.54** | **47.75** | **47.40** | **59.40** | **56.78** | **70.99** | **69.93** |
| SCUT-CTW1500 | LR | 14.63 | 0.53 | 23.55 | 4.76 | 47.23 | 10.18 | 28.92 | 16.39 | 31.16 | 26.19 | 25.82 | 17.71 |
| | BSRGAN | 24.67 | 3.37 | 21.86 | 3.54 | 35.10 | 3.88 | 51.41 | 46.21 | 47.44 | 37.83 | 67.52 | 66.05 |
| | DiffBIR | 24.71 | 4.76 | 23.82 | 5.10 | 30.71 | 4.64 | 62.37 | 54.75 | 61.90 | 49.89 | 71.19 | 63.16 |
| | DiffTSR | 19.77 | 2.95 | 15.98 | 2.86 | 23.69 | 2.90 | 35.39 | 35.49 | 30.59 | 31.88 | 55.83 | 50.43 |
| | InvSR | 29.57 | 2.09 | 26.41 | 2.17 | 34.50 | 2.43 | 57.75 | 29.65 | 55.94 | 29.62 | 69.25 | 40.29 |
| | MARCONet | 0.13 | 0.13 | 0.57 | 0.61 | 0.22 | 0.26 | 33.34 | 33.56 | 16.54 | 16.20 | 28.78 | 28.95 |
| | MARCONet++ | 22.72 | 2.35 | 20.63 | 2.60 | 33.10 | 2.60 | 34.65 | 14.75 | 19.58 | 8.06 | 43.61 | 30.27 |
| | PiSA-SR | 37.46 | 7.61 | 34.14 | 6.92 | 44.11 | 9.43 | 56.31 | 41.77 | 53.05 | 36.75 | 68.19 | 58.95 |
| | Real-ESRGAN | 31.31 | 5.02 | 26.94 | 5.64 | 43.25 | 7.74 | 40.81 | 28.37 | 43.43 | 20.95 | 52.66 | 39.99 |
| | SwinIR | 23.10 | 3.67 | 23.21 | 4.68 | 39.44 | 5.27 | 33.85 | 19.00 | 46.07 | 24.14 | 39.36 | 25.64 |
| | StableSR | 25.55 | 3.33 | 19.95 | 4.43 | 45.86 | 7.49 | 31.04 | 20.93 | 43.61 | 20.92 | 24.92 | 16.62 |
| | SUPIR | 18.26 | 5.43 | 17.61 | 6.26 | 24.37 | 7.00 | 57.35 | 55.46 | 51.68 | 47.02 | 66.96 | 65.55 |
| | TAIR | 33.98 | 10.74 | 29.44 | 9.23 | 41.67 | 12.14 | 65.38 | 63.60 | 47.05 | 36.57 | 67.08 | 66.38 |
| | **GLYPH-SR** | **38.26** | **11.09** | **36.96** | **14.71** | **42.90** | **14.67** | **70.33** | **61.94** | **57.88** | **48.21** | **70.31** | **63.43** |
| CUTE80 | LR | 65.80 | 39.38 | 50.58 | 36.31 | 80.47 | 67.18 | 28.93 | 17.29 | 36.80 | 22.58 | 37.64 | 17.32 |
| | BSRGAN | 73.09 | 55.21 | 56.02 | 46.57 | 83.97 | 71.18 | 44.22 | 42.07 | 55.73 | 54.31 | 69.13 | 67.33 |
| | DiffBIR | 68.88 | 59.56 | 48.82 | 44.71 | 81.84 | 70.53 | 51.04 | 47.53 | 72.64 | 62.09 | 69.06 | 64.62 |
| | DiffTSR | 61.08 | 54.39 | 47.48 | 42.33 | 73.71 | 63.35 | 33.94 | 33.55 | 38.47 | 42.95 | 58.74 | 57.46 |
| | InvSR | 72.46 | 56.42 | 55.62 | 45.18 | 84.75 | 72.46 | 50.30 | 37.66 | 67.78 | 62.43 | 70.66 | 57.69 |
| | MARCONet | 2.31 | 3.07 | 4.56 | 4.56 | 3.82 | 3.82 | 33.58 | 33.66 | 26.69 | 26.69 | 31.06 | 30.88 |
| | MARCONet++ | 69.21 | 50.58 | 54.39 | 45.18 | 81.02 | 71.50 | 31.88 | 21.03 | 34.90 | 22.72 | 54.15 | 44.24 |
| | PiSA-SR | 72.77 | 52.72 | 54.80 | 42.33 | 82.65 | 75.24 | 45.82 | 30.71 | 61.81 | 30.80 | 66.18 | 45.16 |
| | Real-ESRGAN | 73.71 | 59.18 | 58.79 | 49.27 | 84.23 | 74.33 | 38.20 | 35.17 | 48.71 | 36.46 | 60.65 | 56.55 |
| | SwinIR | 73.71 | 52.30 | 55.62 | 45.18 | 82.92 | 72.46 | 31.87 | 22.72 | 59.32 | 40.40 | 47.94 | 39.44 |
| | StableSR | 72.14 | 57.81 | 57.22 | 45.18 | 82.92 | 73.87 | 36.26 | 26.00 | 49.74 | 40.42 | 60.09 | 34.48 |
| | SUPIR | 70.85 | 58.01 | 51.87 | 42.81 | 82.11 | 70.20 | 47.50 | 46.38 | 62.62 | 61.67 | 68.26 | 67.04 |
| | TAIR | 55.21 | 42.81 | 43.77 | 40.87 | 69.87 | 62.20 | 58.25 | 37.11 | 49.76 | 36.84 | 72.06 | 55.06 |
| | **GLYPH-SR** | **73.09** | **63.66** | **55.62** | **45.65** | **85.01** | **73.71** | **49.77** | **47.75** | **65.93** | **65.85** | **69.96** | **68.85** |

| Model | GLYPH-SR | DiffBIR | TAIR | Real-ESRGAN | PiSA-SR | StableSR | SwinIR | BSRGAN | InvSR | SUPIR | DiffTSR | MARCONet | MARCONet++ |
|---|---|---|---|---|---|---|---|---|---|---|---|---|---|
| 1st Rank | 20 | 6 | 4 | 3 | 1 | 1 | 1 | 0 | 0 | 0 | 0 | 0 | 0 |
| 2nd Rank | 8 | 6 | 4 | 2 | 3 | 3 | 0 | 4 | 4 | 3 | 0 | 0 | 0 |

(6 first-place rankings mostly in SR metrics) via hallucination but suffers from low OCR accuracy (e.g., SVT ×4). Conversely, **StableSR** preserves legibility through conservative restoration but yields poor perceptual quality. Notably, DiffTSR, MATRCONet, and MARCONet++ underperform across the board, failing to secure any top rankings, as their exclusive focus on local text features limits their capacity for global scene restoration.

**GLYPH-SR** reconciles these conflicting objectives. As summarized in the ranking statistics at the bottom of Table 1, our model demonstrates overwhelming dominance, securing **20 first-place** and **8 second-place** rankings across 36 metric comparisons. This stands in stark contrast to the nearest competitor, DiffBIR, which achieved only 6 first-place rankings.

Our advantage is most prominent under extreme degradation (×8 scale). On the challenging **SVT ×8** benchmark, GLYPH-SR outperforms the strongest competitor by a remarkable margin of **+15.18 pp** in OpenOCR $F_1$, while maintaining superior perceptual metrics. This confirms that our token-wise guidance effectively prevents both the textual hallucination of GANs and the over-smoothing of generic diffusion models.

Fig. 4 shows that GLYPH-SR successfully harmonizes precise glyph recovery with holistic scene reconstruction. Our model accurately restores the coherent text string "CARROLL STREET BAKERY" without suffering from the severe hallucination (e.g., spurious characters in DiffBIR) or blurring seen

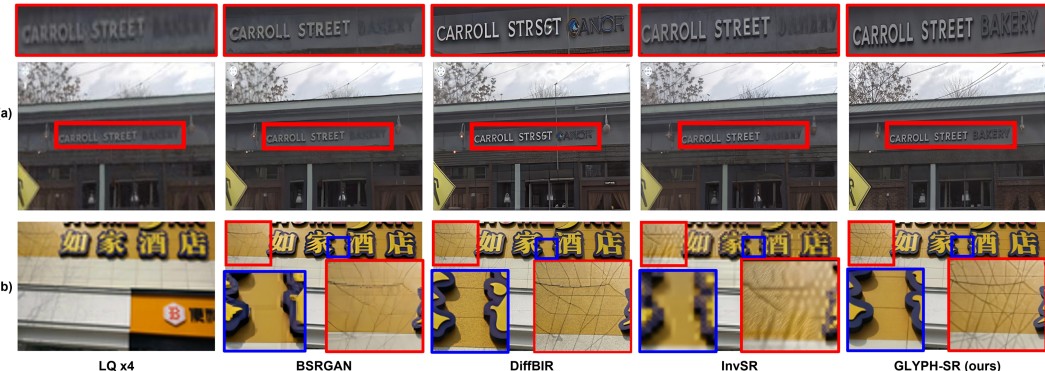

Figure 4: Visualizing the simultaneous achievement of the bi-objective: high-fidelity text restoration and authentic global texture preservation.

in baselines. Crucially, this text fidelity does not compromise the background; GLYPH-SR preserves realistic surface details (e.g., the tiled wall) where competitors introduce mosaic artifacts or fake cracks, confirming that our method avoids the trade-off between text clarity and global image fidelity.

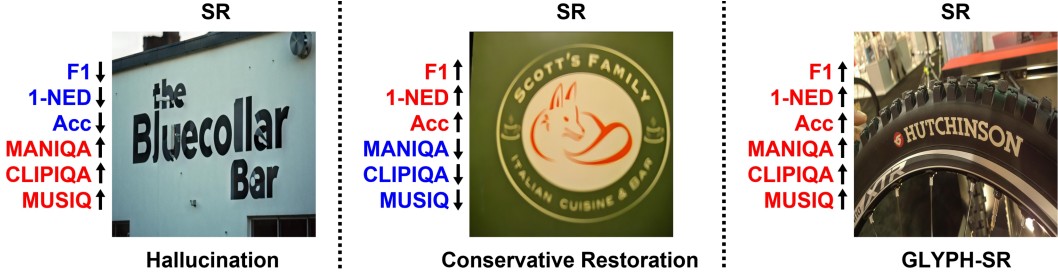

Figure 5: Qualitative examples illustrating the trade-off between SR metrics (e.g., MANIQA, CLIP-IQA, MUSIQ) and OCR metrics ($F_1$, Accuracy) in scene-text images. While some methods improve perceptual SR scores, they may degrade OCR performance, and vice versa.

Fig. 5 concretizes the two failure modes introduced earlier (Fig. 1). The examples on the left illustrate *hallucination*—sharp strokes that alter glyphs, raising IQA scores but breaking legibility. In contrast, those on the right exhibit *conservative restoration*. This issue stems from insufficient SR, a cautious approach to prevent hallucination. While this allows an OCR module to recognize the low-quality text, it results in blurry, low-contrast images with minimal SR gains. By preserving glyph topology while restoring realistic textures, **GLYPH-SR** avoids both pitfalls, yielding images that are both high-quality and OCR-readable. This outcome underscores why evaluations must report SR and OCR metrics jointly for a comprehensive assessment.

Taken together, the results confirm that our method yields a balanced architecture that advances the SOTA by resolving the conflict between text recognition and perceptual SR.

Fig. 6 visually demonstrates how our model uniquely preserves text structure and legibility across severe degradations ($\times 4$ to $\times 8$). Competing methods exhibit clear failure modes. Diffusion models like DiffBIR, despite high perceptual scores, frequently hallucinate incorrect characters (e.g., *'EANK OF ENUNAL'*). Conversely, GAN-based methods like BSRGAN's high contrast produces jagged, geometrically distorted glyphs that harm human readability.

This confirms the trade-off between perceptual quality and OCR accuracy observed in Table 1. Methods that excel in one metric often fail in the other. GLYPH-SR consistently reconciles both objectives, delivering coherent and legible results even at the extreme $\times 8$ scale where other models collapse.

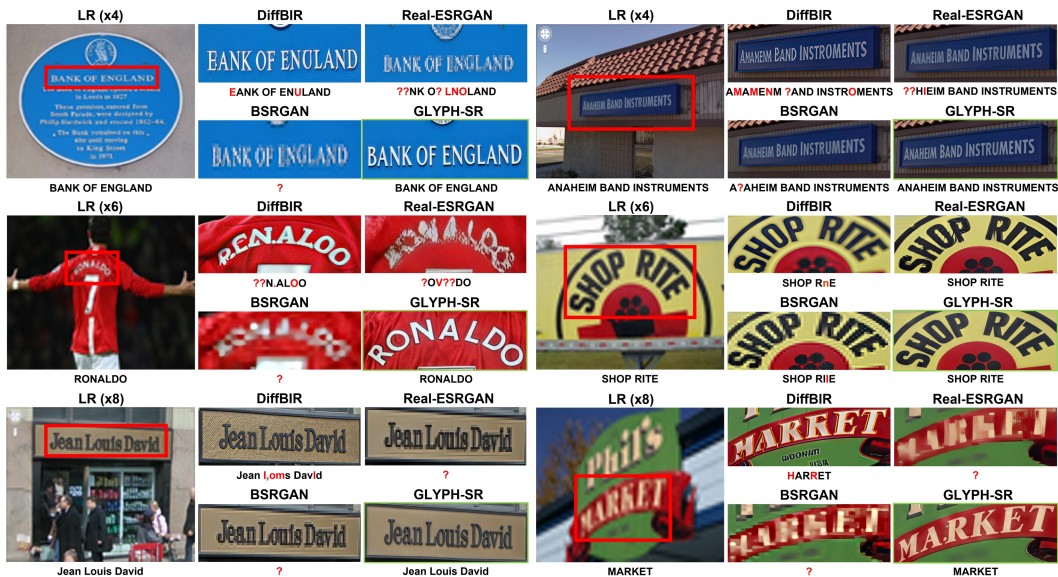

Figure 6: Comparison of SR results against different methods (DiffBIR, Real-ESRGAN, BSRGAN, and GLYPH-SR) on various degraded LR images.

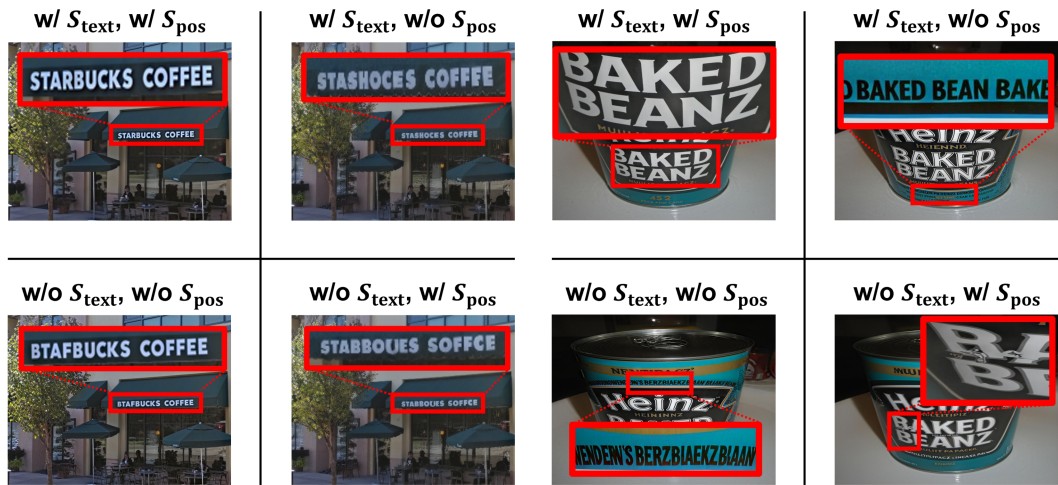

Figure 7: Four prompt settings using combinations of texts ($\mathcal{S}_{\text{text}}$) and its spatial positions ($\mathcal{S}_{\text{pos}}$).

### 4.2.1 ABLATION STUDIES

Fig. 7 shows the effect of selectively removing the two of guidance used by GLYPH-SR: (i) the OCR string $\mathcal{S}_{\text{text}}$ and (ii) its spatial positions $\mathcal{S}_{\text{pos}}$. We evaluate four combinations—*both*, *text-only*, *position-only* and *none*.

1) Full guidance ( $\mathcal{S}_{\text{text}} + \mathcal{S}_{\text{pos}}$ ): The top-left quadrants reconstruct the text pattern without distortions, retaining stroke width, inter-letter spacing, and global geometry.

2) Text-only guidance ( $\mathcal{S}_{\text{text}} / \mathcal{S}_{\text{pos}}$ ): When positional guidance is removed, the model hallucinates irregular kerning and warped baselines (e.g. "*STASHOES COFFEE*"), indicating that semantics alone cannot anchor glyph layout.

3) Position-only guidance ($\mathcal{S}_{\text{text}} / \mathcal{S}_{\text{pos}}$ ): Conversely, supplying bounding boxes but no textual content yields partial or incorrect spellings ("*STABHOUES SOFFCE*"), showing that location cues without semantics lead to character-level ambiguity.

4) No guidance ($\mathcal{S}_{\text{text}}+\mathcal{S}_{\text{pos}}$): Removing both priors produces the worst outcomes—severe hallucinations and geometric distortions reminiscent of generic diffusion SR.

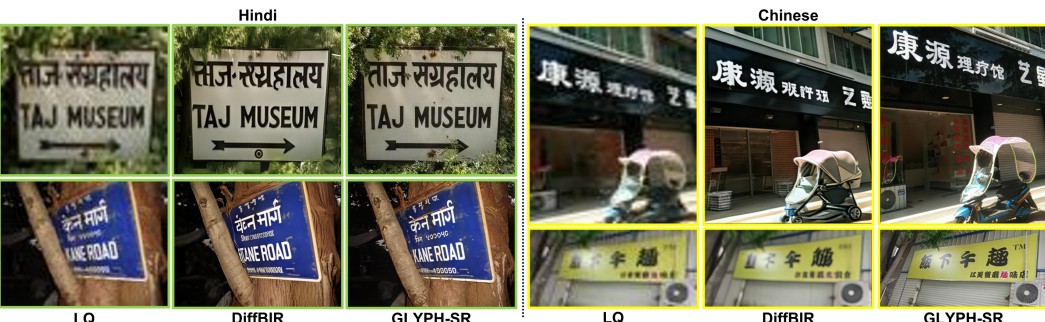

Figure 8: Visual comparison of DiffBIR and GLYPH-SR on non-Latin scripts (Chinese and Hindi).

To validate cross-lingual generalization, we evaluated GLYPH-SR on Chinese (LSVT Sun et al. (2019)) and Devanagari (IndicSTR Mathew et al. (2017)) scripts by simply replacing the default English-centric guider with a multilingual VLM (Gemini 2.5 Flash). As shown in Fig. 8, the GYLPH-SR recovers complex glyph geometries and stroke patterns that differ significantly from Latin scripts. This confirms that our TS-ControlNet learns to correct generic structural degradation rather than language-specific rules. Consequently, our modular design—decoupling semantic guidance from the frozen generative backbone—enables effective restoration across diverse languages without the need to retrain the core model.

## 5 CONCLUSIONS

Super-resolution research has traditionally prioritized perceptual quality, often neglecting a critical aspect of text-rich scenes: legibility. This creates a persistent gap where models produce sharp-looking images that still cannot be read correctly, as text is underweighted by standard SR objectives. To resolve this, GLYPH-SR reframes the task as a bi-objective problem that optimizes both visual realism and text legibility. We introduce a practical recipe featuring a VLM-guided diffusion model with a dual-branch TS-ControlNet, which fuses spatial OCR cues and a global caption. To properly evaluate this balance, we provide a factorized synthetic corpus and a dual-axis protocol pairing OCR $F_1$ with perceptual IQA metrics. On challenging benchmarks (SVT, SCUT-CTW1500, CUTE80 at ×4/×8), GLYPH-SR improves OCR $F_1$ by up to +15.18 pp over strong baselines while maintaining top-tier perceptual quality. Future work will explore multilingual scripts, stronger geometric priors, and tighter integration with end-to-end recognition systems.

### REPRODUCIBILITY STATEMENT

We are committed to the reproducibility of our work. The complete source code, pretrained models, synthetic data generation scripts, and evaluation suite for our **GLYPH-SR** framework are provided as supplementary material, with direct links available in Appendix B. Details of the model architectures, pre-trained backbones, and key hyper-parameters are described in Section 4.1 and extensively in Appendix B, which also specifies the hardware and software environment used for all experiments. Our dual-axis evaluation protocol, including all benchmark datasets (SVT, SCUT-CTW1500, CUTE80) and the specific OCR and perceptual metrics, is documented in Section 4.1 and Appendix A.1. The core components of our method, including the Text-SR Fusion ControlNet, condition decomposition, and the ping-pong scheduler, are detailed in Section 3. The data generation and fine-tuning workflow, serving as a practical guide, is outlined in Appendix B. Extended experimental results, comprehensive ablation studies, sensitivity analyses, and a discussion of the computational footprint are reported in Appendix C to ensure full transparency of our findings. These resources provide a comprehensive toolkit for the faithful reproduction and independent verification of our results.

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
