# Supplementary Material

TABLE OF CONTENTS

# A METRIC-BASED EVALUATION

## A.1 EVALUATION METRICS

We evaluate our method along two primary axes: (1) the *semantic integrity* of textual content, and (2) the *perceptual quality* of the reconstructed images. Accordingly, we organize the metrics into two groups.

**OCR Metrics.** To assess text restoration performance, we report:

- $F_1$ **score**, **Precision**, **Recall** and **Accuracy** (↑), : character-level measures of OCR correctness; higher is better.
- **Normalized Edit Distance (1-NED)** (↑): inverse of edit distance, scaled to [0, 100]; higher values indicate closer agreement with the ground truth.

**Image-Quality Metrics.** For perceptual fidelity, we adopt:

- **Peak Signal-to-Noise Ratio (PSNR)** (↑): log-scaled pixel-level similarity to the reference image.
- **Structural Similarity Index (SSIM)** (↑): evaluates luminance, contrast, and structural consistency in line with human perception, scaled to 0–100.
- **Learned Perceptual Image Patch Similarity (LPIPS Zhang et al. (2018a))** (↓): deep-feature distance reflecting perceptual differences, scaled to 0–100.
- **Multi-Dimension Attention Network for No-Reference IQA (MANIQA Yang et al. (2022))** (↑): no-reference quality score based on attention-driven features, scaled to 0–100.
- **CLIP-based Image Quality Assessment (CLIP-IQA Wang et al. (2023))** (↑): semantic fidelity metric leveraging CLIP embeddings, scaled to 0–100.
- **Multi-Scale Image Quality Transformer (MUSIQ Ke et al. (2021))** (↑): transformer-based no-reference IQA that aggregates multi-resolution cues.

## A.2 MISALIGNMENT BETWEEN METRICS AND HUMAN PERCEPTION

SR papers still default to **PSNR**, **SSIM**, and **LPIPS**. Although convenient, these scores often drift from what people actually perceive—especially when the low-resolution input is heavily degraded. Fig. 9 offers four counter-examples that highlight three recurring failure modes.

**Perceptual vs. Semantic Fidelity.** In Figure 9, the "HOMER BREWING COMPANY" sign is reconstructed cleanly by GLYPH-SR yet receives lower PSNR and SSIM than Real-ESRGAN, whose output contains aliasing and hallucinated glyphs.

**Metrics Can Be Misleading.** Across multiple benchmarks we frequently observe visually superior outputs that score lower on PSNR/SSIM/LPIPS (see red vs. blue values in Figure 9). This misalignment—echoed by prior studies Blau & Michaeli (2018); Jinjin et al. (2020); Gu et al. (2022); Yu et al. (2024a)—underscores the danger of metric-only evaluation. For text-aware SR, side-by-side inspection or user studies remain indispensable.

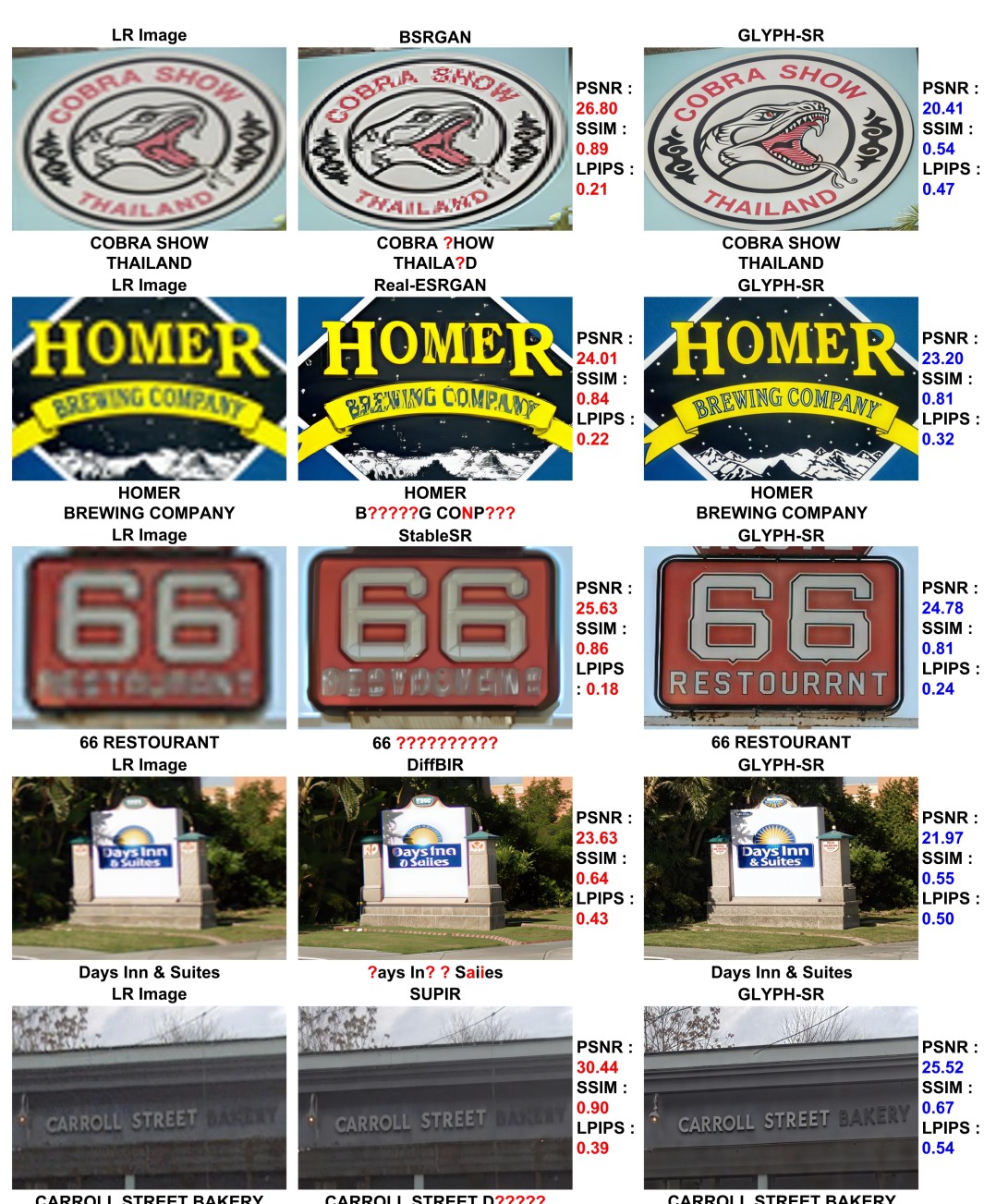

Figure 9: Each triplet shows (left) the input LR image, (middle) a strong baseline, and (right) GLYPH-SR. Despite GLYPH-SR producing visibly sharper text, its PSNR/SSIM/LPIPS scores (blue) are often lower than those of the baseline (red). The gap exposes a growing consensus: traditional metrics alone do not capture human perception of text-laden imagery.

# B EXPERIMENT DETAILS

## B.1 REPRODUCIBILITY STATEMENT

**Synth Dataset:**
https://drive.google.com/drive/folders/1eYMvZQq-93okI2v1YldXLPHDyc3kuvdu?usp=drive_link

**Pretrained Model:**
https://drive.google.com/drive/folders/1hrZ5jRbVLcRSFpbL-uPxe9iLddylAFgk?usp=drive_link

**Code:**
https://drive.google.com/drive/folders/1A75nhOQEG1hcEhzUJxO75X8LfT07lR3K?usp=drive_link

**Results:**
https://drive.google.com/drive/folders/1CArNuM0AI50z3TGsR66u218RLV5UdHYa?usp=drive_link

**Data Generation & Fine-Tuning Workflow**

1. **Stage 1 – Scene Description Extraction**
   `dataset_generater/make_dataset_get_desc.py`
   `./datasets/descriptions/` containing: *{id, image_path, ocr_text, caption}*.

2. **Stage 2 – Augmented Prompt Synthesis**
   `third_party/make_dataset_with_nunchaku/`
   `make_dataset_with_augmentation.py`
   Invokes the *Nunchaku* augmentation engine to expand each record with synthetic corruptions (blur, noise, JPEG artifacts) and with diversity-enhanced textual prompts. The output is a paired folder structure: `./datasets/aug/{hq, lq}`.

3. **Stage 3 – Negative/HQ Pairing**
   `dataset_generater/make_dataset_Neg_HQ.py`
   Generates explicit (LQ, HQ) pairs and the associated prompt metadata required by GLYPH-SR. Final training files are placed under `./datasets/final/`.

4. **Stage 4 – Fine-Tuning**
   `train_GLYPH_SR.py`

   ```
   python3 train_GLYPH_SR.py \
           --data_root ./datasets/ \
           --cfg       GLYPH-SR/model_configs/model_config.yaml
   ```

**Inference Workflow**

1. **Create the checkpoint directory.**
   Download every model file from the *Pre-trained Checkpoints* link and place them in a newly created folder named `CKPT_PTH` at the project root.

2. **Patch all path references.**
   Edit the three files listed below so that each points to the new directory, e.g. `CKPT_PTH/<checkpoint_name>.pth`:
   - `GLYPH-SR/model_configs/model_config.yaml`
   - `GLYPH-SR/run_GLYPH_SR.py`
   - `GLYPH-SR/CKPT_PTH.py`

3. **Run command.**
   Verify correct loading by launching a single-image run:

   ```
   python3 run_GLYPH_SR.py --img_path ./image.jpg
   ```

Successful execution confirms that all checkpoints are discovered and that GLYPH-SR is ready for inference.

## B.2 SETUP

All experiments were conducted on a workstation equipped with three NVIDIA RTX 6000 Ada GPUs (48 GB each), all utilized concurrently for training and inference, an Intel Xeon W9-3475X CPU (36 cores, 72 threads), and 256GB RAM. The system runs Ubuntu 24.04.2 LTS and uses a 3.7TB NVMe SSD for storage. The models were implemented in PyTorch 2.5.1 with CUDA 11.8.

### B.2.1 EVALUATION DATASETS

**ICDAR2017 (International Conference on Document Analysis and Recognition).**  The IC-DAR2017 Robust Reading Challenge dataset consists of scene-text images designed to test text detection and recognition systems under real-world conditions. It includes both high- and low-quality images that exhibit various degradations such as blur, low resolution, and noise. This diversity makes it well-suited for fine-tuning vision-language models to enhance OCR robustness. In our pipeline, ICDAR2017 is used to fine-tune LLaVA-NeXT, enabling it to better handle degraded scene-text images and produce more accurate token-level guidance.

**SCUT-CTW1500 (Curved Text in the Wild).**  SCUT-CTW1500 is a large-scale scene-text detection benchmark featuring 1,500 images with over 10,000 annotated curved text instances. The dataset includes a wide variety of natural scenes such as street views, signboards, and shop names, with text appearing in arbitrary orientations, lengths, and curvature. It is especially known for its high diversity in text shape and layout, which makes it well-suited for evaluating the robustness of text detection and SR models in processing long and curved text lines. SCUT-CTW1500 is widely used for benchmarking models designed to process irregular and multi-oriented scene-text under real-world conditions.

**CUTE80 (Curve Text).**  CUTE80 is a compact yet challenging dataset containing 80 high-resolution images, specifically curated to evaluate curved text detection and recognition systems. The dataset features a range of naturally curved and perspective-distorted text instances embedded in complex backgrounds such as logos, signs, and posters. Despite its small size, CUTE80 is frequently used in literature to benchmark the generalization ability of text-focused models on non-horizontal and non-linear text structures. Its emphasis on difficult geometric deformations makes it a useful supplement to larger datasets for testing text-specific visual models under challenging conditions.

**SVT (Street View Text).**  SVT is a benchmark dataset collected from Google Street View, consisting of 647 images with approximately 2,000 annotated text instances. It features naturally occurring scene-text with various distortions, backgrounds, lighting conditions, and orientations. Despite its relatively small size, SVT is widely used in the literature for benchmarking the performance of OCR and text SR models under real-world conditions. Its challenging scenarios make it suitable for evaluating model generalization and robustness in unconstrained environments.

### B.2.2 PRE-TRAINED MODELS

**LLaVA-NeXT.**  We employ **LLaVA-NeXT** Liu et al. (2024) as the vision–language front-end that extracts semantic context from low-resolution inputs. LLaVA-NeXT couples a CLIP-ViT visual encoder with a 7-B parameter LLM and is instruction-tuned on a large multimodal corpus, yielding state-of-the-art performance in fine-grained grounding, captioning, and region-level reasoning. Within our pipeline it automatically produces (i) image-level captions (*IMG prompts*) and (ii) spatially aligned OCR strings (*OCR prompts*); both streams are fed as high-level conditions to the diffusion backbone.

**JuggernautXL (SDXL-based).**  For image generation we adopt **JuggernautXL**, a publicly released checkpoint built on *SDXL-base 1.0* and further fine-tuned for improved sharpness and color fidelity. The underlying SDXL architecture is trained on billions of image–text pairs and natively supports $1024 \times 1024$ resolution.

### B.2.3 KEY HYPER-PARAMETERS

- **Vision–Language Encoder.** A frozen LLaVA-NeXT produces $2\,048$-dimensional multi-modal embeddings that act as cross-attention keys; because the encoder is not fine-tuned, it adds zero trainable parameters.

- **First Stage (VAE).** A $256 \times 256$ auto-encoder (4 latent channels, $4\times$ down-sampling) maps RGB images to a $64 \times 64 \times 4$ latent grid.

- **Denoising and Sampling.** We use the standard $1\,000$-step DDPM schedule wrapped by RESTORE-EDM sampling (default: 50 inference steps, classifier-free guidance scale annealed from 7.5 to 4.0).

### B.2.4 SYNTHETIC TRAINING DATA

To train the TS-ControlNet, we curate a purpose-built synthetic dataset with four mutually exclusive partitions: *Positive/High-Quality*, *Positive/Low-Quality*, *Negative/High-Quality*, and *Negative/Low-Quality*. Each split is created by selectively degrading either global content or localized glyph regions while keeping spatial layout and annotations intact. This design lets the network disentangle text-specific cues from general image priors.

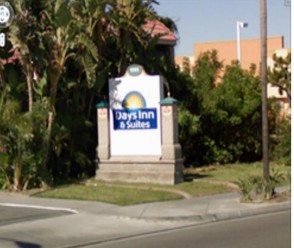

"**id**": "/SVT_image_x4/00_18.jpg"

"**OCR**": "Days Inn & Suites"

"**prompt**": "The image depicts a street scene with a focus on a sign for a hotel named **Days Inn & Suites.** ….. The image has a casual, everyday quality to it, likely intended to show the location of the hotel for travelers or passersby."

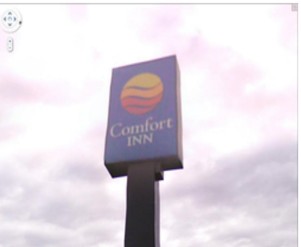

"**id**": "/SVT_image_x4/00_19.jpg"

"**OCR**": "Comfort Inn"

"**prompt**": "The image shows a sign for a hotel or motel named **Comfort Inn.** The sign is rectangular with rounded corners and is mounted on a vertical pole. ….. The focus is on the sign, and the image is taken from a slightly lower angle, which makes the sign stand out against the sky."

Figure 10: Step 1: JSONL metadata—`id`, OCR text, and scene prompt—generated by LLAVA-NEXT.

**Step 1: Prompt-Metadata Extraction.** Using a pretrained LLAVA-NEXT encoder and YAML-defined prompt templates, we batch-process scene-text images and record three fields in `JSONL`: image `id`, OCR text, and a scene-level `prompt`. Figure 10 illustrates the resulting metadata, produced by `make_dataset_get_desc.py`.

**Step 2: Stylistic Augmentation.** We prepend each prompt with a style token (e.g., *sunset glow*, *cinematic bokeh*) via `make_dataset_with_augmentation.py`. The enriched prompts drive a Flux-based diffusion model equipped with ControlNet and LoRA modules to generate visually diverse high-quality samples (Fig. 11).

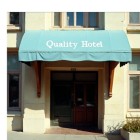 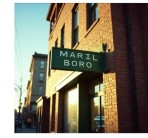

**"prompt"**: "Fujifilm Pro 400H color palette , The text **Quality Hotel** is displayed in white capital letters on a light blue awning above the entrance to the hotel building. The awning is supported by wooden brackets and casts a slight shadow on the facade of the building, creating a clear contrast against the beige stone wall. Flanking the entrance are two rectangular, realistic photograph, 35 mm film style, soft natural lighting."

**"prompt"**: "golden hour warmth , The text **MARIL BORO** appears as a green, block-capital sign affixed to the storefront of a brick building, bathed by sunlight, and surrounded by a clear blue sky, realistic photograph, 35 mm film style, soft natural lighting."

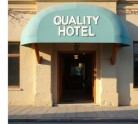 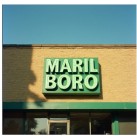

**"prompt"**: "sunset glow , The text **Quality Hotel** is displayed in white capital letters on a light blue awning above the entrance to the hotel building. The awning is supported by wooden brackets and casts a slight shadow on the facade of the building, creating a clear contrast against the beige stone wall. Flanking the entrance are two rectangular, realistic photograph, 35 mm film style, soft natural lighting."

**"prompt"**: "Fujifilm Pro 400H color palette , The text **MARIL BORO** appears as a green, block-capital sign affixed to the storefront of a brick building, bathed by sunlight, and surrounded by a clear blue sky, realistic photograph, 35 mm film style, soft natural lighting."

Figure 11: Step 2: Prompt augmentation with stylistic keywords to boost visual diversity.

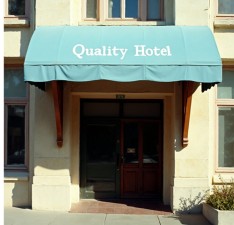 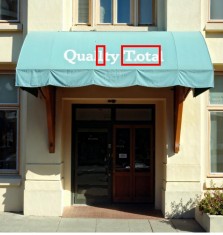 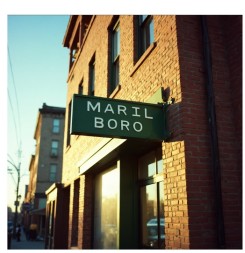 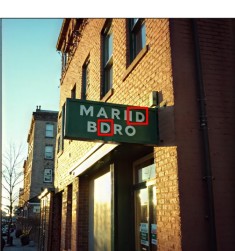

**Positive High Quality Image**    **Negative High Quality Image**    **Positive High Quality Image**    **Negative High Quality Image**

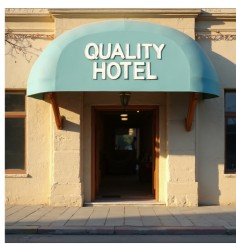 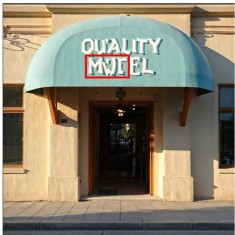 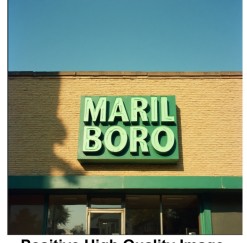 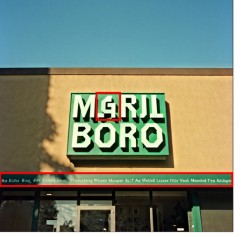

**Positive High Quality Image**    **Negative High Quality Image**    **Positive High Quality Image**    **Negative High Quality Image**

Figure 12: Step 3: Positive vs. intentionally corrupted (negative) high-quality pairs.

**Step 3: Negative High-Quality Pairs.** The `make_dataset_Neg_HQ.py` script corrupts text regions at the glyph level while leaving global detail untouched, yielding hard negative examples. Corruptions are verified with the SUPIR pipeline Yu et al. (2024b)(Fig. 12).

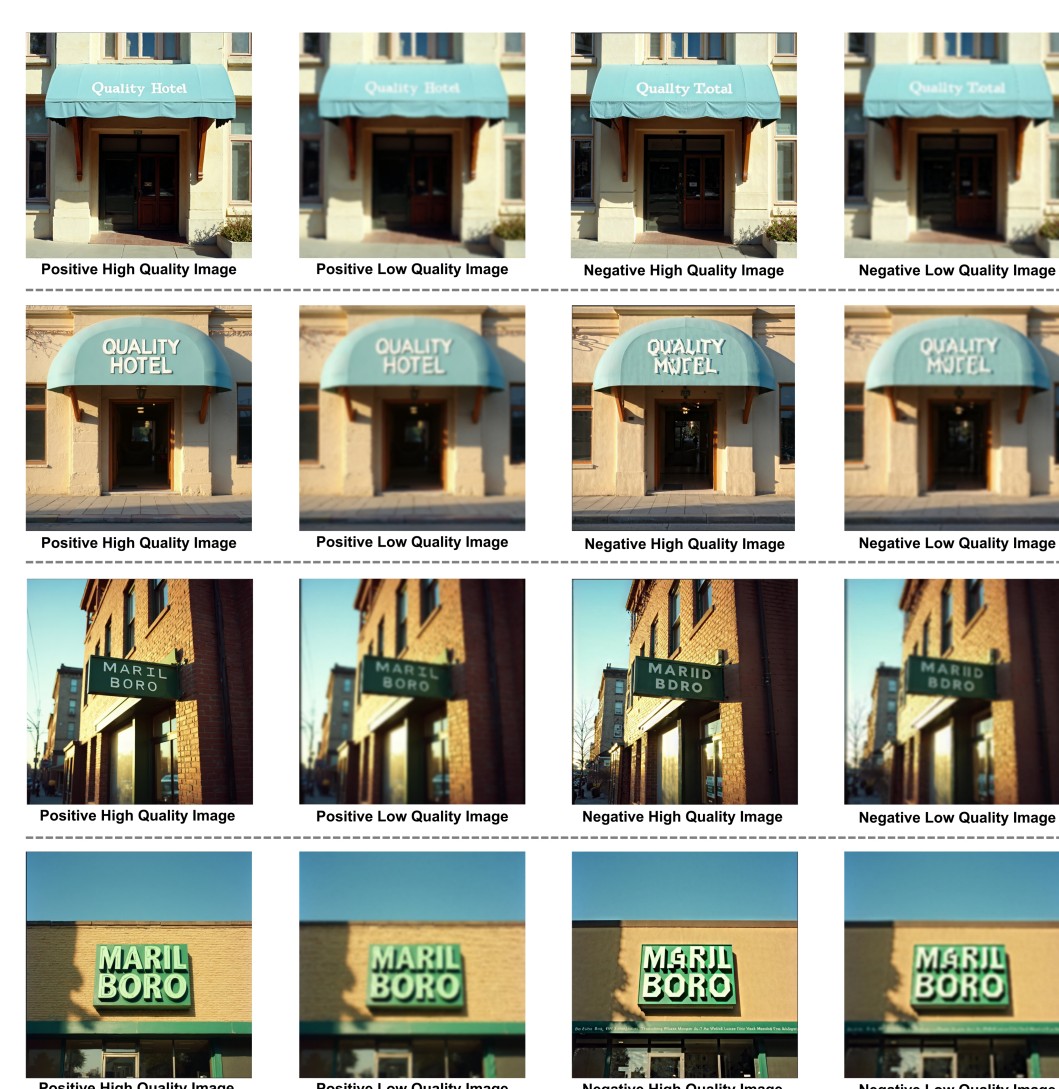

Figure 13: Step 4: Four synthetic subsets used for text-aware SR training.

**Step 4: Final Dataset Assembly.** All positive and negative images are merged into the four target splits shown in Fig. 13.

# C EXPERIMENT RESULTS

## C.1 COMPARE CHARACTER GENERATION TO OTHER MODELS.

We compare our model's character generation ability against standard OCR models across difficulty levels. The results in Table 2 show significant improvements, especially under hard conditions. For evaluation, LLaVA-NeXT and our model were prompted using the following instructions: *"Please perform OCR on this image."* Additionally, both predicted and ground truth texts were normalized by removing non-alphabetic characters and ignoring case sensitivity before computing the metrics.

Fig. 18 provides a qualitative comparison between GLYPH-SR and baselines at magnification factors of $\times 4$, $\times 8$, and an extreme $\times 16$. GLYPH-SR continuously reconstructs glyph outlines, stroke widths, and kerning while remaining true to the underlying truth, while harmonizing color and brightness with the surrounding background area. This visual evidence corroborates the quantitative gap observed in Table 1: models that optimize solely for perceptual metrics (e.g. DiffBIR, Real-ESRGAN) or for edge contrast fall short on OCR fidelity once the scale factor exceeds $\times 8$.

## C.2 ABLATION STUDY

In this ablation, we test an alternative to our binary ping-pong policy: several static **"Mixing"** policies. These policies use a constant mixing ratio $\lambda_t = C$ (where $C \in \{0.1, 0.3, 0.5, 0.7, 0.9\}$) to blend the text and image guidance at every step, rather than alternating. The results in Table 4 demonstrate that this static mixing leads to an unstable and sub-optimal trade-off.

As detailed in Table 5, increasing $w$ generally improves perceptual metrics (e.g., CLIP-IQA, MUSIQ) by strengthening the generative prior. However, we observed a distinct trade-off where excessive guidance ($w > 7.5$) leads to a degradation in OCR performance.

The control scale modulates the influence of our Text-SR Fusion ControlNet. As presented in Table 6, we identified severe failure modes associated with high values of $s_{\text{CTRL}}$. When the scale exceeds 1.0, the strong injection of control features disrupts the natural image statistics of the diffusion backbone, leading to over-conditioning.

## C.3 SENSITIVITY TO UPSTREAM VLM/OCR ERRORS

We assess how errors in upstream text guidance (OCR/VLM) propagate to GLYPH-SR. Because our method deliberately conditions on token-level strings and locations, corrupted guidance could degrade both readability and overall perceptual quality. We simulate three error modes and measure their impact on OCR and IQA metrics.

1. **Random Character Corruption:** Replace $n\% \in \{30, 50, 90\}$ of characters in the OCR string with uniformly sampled alternatives (random noise).

2. **Plausible Character Swaps ("Swap"):** Systematically replace characters with visually confusable counterparts from a curated set (e.g., O$\leftrightarrow$0, I$\leftrightarrow$1, T$\leftrightarrow$7).

3. **Missed Detections ("Drop"):** Remove a portion of OCR-recognized characters to emulate detection/recognition failures.

Table 7 reports OpenOCR/GOT-OCR $F_1$ and MANIQA/CLIP-IQA. Parentheses show absolute changes w.r.t. the uncorrupted baseline.

All error modes substantially hurt both axes: readability (*OpenOCR/GOT-OCR* $F_1$) and perceived image quality (MANIQA/CLIP-IQA). Even moderate noise (50%) reduces OpenOCR $F_1$ by 16.79,pp and MANIQA by 16.62 points. Plausible swaps and missed detections also trigger large drops, indicating that quantity (how many tokens are wrong), nature (plausible vs. random), and absence (drops) all impair glyph integrity and global appearance. This validates our design choice to use a strong, LR-aware OCR/VLM and to treat guidance quality as a first-order factor in text-aware SR.

To investigate the trade-off mechanism empirically, we visualized the internal noise prediction maps $\epsilon_\theta$ across diffusion timesteps, as shown in Fig. 22. We compared the feature map evolution under

Table 2: Quantitative comparison of OCR performance on images degraded by various factors and restored using six SR models, evaluated across three benchmark datasets and three OCR systems. Red and blue denote the best and second-best results, respectively.

| | OpenOCR | | | | | GOT-OCR | | | | | LLaVA-NeXT | | | | |
|---|---|---|---|---|---|---|---|---|---|---|---|---|---|---|---|
| | Precision | Recall | $F_1$ score | 1-NED | Accuracy | Precision | Recall | $F_1$ score | 1-NED | Accuracy | Precision | Recall | $F_1$ score | 1-NED | Accuracy |
| **SVT(x4)** | | | | | | | | | | | | | | | |
| LR | 61.41 | 40.28 | 48.65 | 26.37 | 32.14 | 75.70 | 59.91 | 66.89 | 34.60 | 50.25 | 83.27 | 60.50 | 70.08 | 13.50 | 53.94 |
| HR | 74.73 | 87.86 | 80.76 | 27.69 | 67.73 | 81.18 | 82.97 | 82.06 | 34.80 | 69.58 | 87.68 | 82.14 | 84.82 | 17.77 | 73.65 |
| BSRGAN | 57.03 | 51.19 | 53.96 | 28.04 | 36.95 | 69.63 | 50.68 | 58.66 | 31.29 | 41.50 | 84.60 | 57.55 | 68.50 | 15.82 | 52.09 |
| DiffBIR | 41.49 | 36.31 | 38.73 | 20.21 | 24.01 | 48.34 | 37.65 | 42.33 | 24.47 | 26.85 | 63.20 | 35.16 | 45.19 | 13.34 | 29.19 |
| DiffTSR | 39.55 | 12.81 | 19.35 | 14.27 | 10.71 | 51.76 | 14.39 | 22.51 | 15.97 | 12.68 | 78.98 | 17.94 | 29.23 | 5.49 | 17.12 |
| InvSR | 55.56 | 60.22 | 57.79 | 27.53 | 40.64 | 65.44 | 57.05 | 60.96 | 31.65 | 43.84 | 78.67 | 55.38 | 65.00 | 15.95 | 48.15 |
| MARCONet | 0.00 | 0.00 | 0.00 | 0.03 | 0.00 | 12.50 | 0.12 | 0.25 | 2.02 | 0.12 | 0.00 | 0.00 | 0.00 | 0.08 | 0.00 |
| MARCONet++ | 55.08 | 45.85 | 50.05 | 27.74 | 33.37 | 68.58 | 53.14 | 59.88 | 32.49 | 42.73 | 81.76 | 55.19 | 65.90 | 15.01 | 49.14 |
| PiSA-SR | 60.16 | 66.79 | 63.30 | 26.71 | 46.31 | 66.84 | 63.70 | 65.23 | 33.74 | 48.40 | 83.20 | 57.14 | 67.75 | 15.44 | 51.23 |
| Real-ESRGAN | 59.41 | 58.89 | 59.15 | 30.16 | 42.00 | 75.05 | 61.04 | 67.32 | 33.50 | 50.74 | 83.70 | 63.99 | 72.53 | 16.25 | 56.90 |
| SwinIR | 58.21 | 51.43 | 54.61 | 28.31 | 37.56 | 72.41 | 56.59 | 63.53 | 34.02 | 46.55 | 83.54 | 64.86 | 73.03 | 16.22 | 57.51 |
| StableSR | 62.08 | 57.83 | 59.88 | 30.32 | 42.73 | 73.79 | 56.13 | 63.76 | 34.71 | 46.80 | 84.70 | 65.56 | 73.91 | 16.81 | 58.62 |
| SUPIR | 58.16 | 58.67 | 58.41 | 20.17 | 41.26 | 64.54 | 59.48 | 61.90 | 26.81 | 44.83 | 74.54 | 53.28 | 62.14 | 15.86 | 45.07 |
| TAIR | 42.11 | 20.13 | 27.23 | 20.81 | 15.76 | 46.60 | 22.26 | 30.13 | 23.61 | 17.73 | 63.71 | 21.88 | 32.58 | 10.04 | 19.46 |
| GLYPH-SR (ours) | 61.33 | 75.14 | 67.54 | 22.17 | 50.99 | 68.07 | 75.79 | 71.72 | 28.37 | 55.91 | 79.22 | 68.07 | 73.22 | 19.49 | 57.76 |
| **SCUT-CTW1500(x4)** | | | | | | | | | | | | | | | |
| LR | 49.86 | 8.57 | 14.63 | 21.56 | 7.89 | 61.44 | 14.57 | 23.55 | 24.51 | 13.35 | 80.62 | 33.40 | 47.23 | 21.73 | 30.91 |
| HR | 72.57 | 68.73 | 70.59 | 56.36 | 54.55 | 75.74 | 65.33 | 70.15 | 52.17 | 54.02 | 87.67 | 69.89 | 77.77 | 35.27 | 63.63 |
| BSRGAN | 46.41 | 16.80 | 24.67 | 29.86 | 14.07 | 56.71 | 13.54 | 21.86 | 23.70 | 12.27 | 76.77 | 15.34 | 25.56 | 17.25 | 21.28 |
| DiffBIR | 38.18 | 18.26 | 24.71 | 33.85 | 14.09 | 36.43 | 17.70 | 23.82 | 30.71 | 13.52 | 54.93 | 21.31 | 30.71 | 20.94 | 18.14 |
| DiffTSR | 45.86 | 12.60 | 19.77 | 25.83 | 10.97 | 50.84 | 9.48 | 15.98 | 18.64 | 8.69 | 72.82 | 14.14 | 23.69 | 11.30 | 13.43 |
| InvSR | 45.37 | 21.93 | 29.57 | 34.39 | 17.35 | 47.40 | 18.31 | 26.41 | 28.17 | 15.22 | 66.15 | 23.33 | 34.50 | 18.34 | 20.84 |
| MARCONet | 17.65 | 0.07 | 0.13 | 0.71 | 0.07 | 13.83 | 0.29 | 0.57 | 2.55 | 0.29 | 41.67 | 0.11 | 0.22 | 0.20 | 0.11 |
| MARCONet++ | 47.02 | 14.98 | 22.72 | 27.05 | 12.82 | 54.82 | 12.70 | 20.63 | 22.17 | 11.50 | 74.92 | 21.24 | 33.10 | 15.93 | 19.83 |
| PiSA-SR | 49.11 | 30.27 | 37.46 | 40.32 | 23.04 | 56.25 | 24.50 | 34.14 | 33.47 | 20.58 | 71.18 | 31.96 | 44.11 | 23.23 | 28.30 |
| Real-ESRGAN | 52.95 | 22.22 | 31.31 | 33.69 | 18.56 | 59.50 | 17.41 | 26.94 | 26.49 | 15.57 | 79.94 | 29.65 | 43.25 | 20.12 | 27.59 |
| SwinIR | 49.75 | 15.05 | 23.10 | 28.11 | 13.06 | 59.40 | 14.42 | 23.21 | 24.64 | 13.13 | 80.24 | 26.14 | 39.44 | 17.58 | 24.56 |
| StableSR | 53.58 | 16.77 | 25.55 | 30.02 | 14.64 | 57.67 | 12.06 | 19.95 | 22.18 | 11.08 | 79.31 | 32.25 | 45.86 | 21.07 | 29.75 |
| SUPIR | 39.95 | 11.84 | 18.26 | 25.73 | 10.05 | 45.16 | 10.93 | 17.61 | 21.40 | 9.65 | 62.60 | 15.13 | 24.37 | 14.32 | 13.87 |
| TAIR | 46.43 | 26.80 | 33.98 | 38.09 | 20.47 | 50.58 | 20.76 | 29.44 | 32.51 | 17.26 | 66.76 | 30.29 | 41.67 | 23.32 | 26.32 |
| GLYPH-SR (ours) | 48.82 | 31.46 | 38.26 | 37.75 | 23.66 | 47.45 | 30.27 | 36.96 | 36.09 | 22.67 | 63.59 | 32.37 | 42.90 | 25.86 | 27.31 |
| **CUTE(x4)** | | | | | | | | | | | | | | | |
| LR | 72.41 | 60.29 | 65.80 | 51.93 | 49.03 | 75.66 | 52.27 | 50.58 | 50.32 | 44.75 | 93.72 | 79.51 | 80.47 | 38.43 | 75.49 |
| HR | 77.78 | 80.38 | 79.06 | 51.61 | 65.37 | 75.33 | 51.36 | 61.08 | 45.35 | 43.97 | 95.17 | 79.76 | 86.78 | 37.54 | 76.65 |
| BSRGAN | 68.84 | 77.89 | 73.09 | 54.63 | 57.59 | 69.44 | 46.95 | 56.02 | 45.37 | 38.91 | 92.54 | 76.86 | 83.97 | 39.00 | 72.37 |
| DiffBIR | 64.90 | 73.37 | 68.88 | 48.01 | 52.53 | 61.48 | 40.49 | 48.82 | 43.45 | 32.30 | 88.12 | 76.39 | 81.84 | 38.53 | 69.26 |
| DiffTSR | 64.94 | 57.65 | 61.08 | 51.95 | 43.97 | 67.80 | 36.53 | 47.48 | 45.54 | 31.13 | 92.59 | 61.22 | 73.71 | 30.66 | 58.37 |
| InvSR | 70.19 | 74.87 | 72.46 | 53.54 | 56.81 | 72.79 | 45.00 | 55.62 | 43.91 | 38.52 | 90.87 | 79.41 | 84.75 | 37.15 | 73.54 |
| MARCONet | 100.0 | 1.17 | 2.31 | 0.78 | 1.17 | 54.55 | 2.38 | 4.56 | 3.74 | 2.33 | 100.00 | 1.95 | 3.82 | 0.96 | 1.95 |
| MARCONet++ | 68.34 | 70.10 | 69.21 | 54.17 | 52.92 | 71.64 | 43.84 | 54.39 | 44.30 | 37.35 | 93.09 | 71.72 | 81.02 | 39.79 | 68.09 |
| PiSA-SR | 71.36 | 74.24 | 72.77 | 50.28 | 57.20 | 70.29 | 44.91 | 54.80 | 42.70 | 37.74 | 93.30 | 74.18 | 82.65 | 38.00 | 70.43 |
| Real-ESRGAN | 71.43 | 76.14 | 73.71 | 53.32 | 58.37 | 71.81 | 49.77 | 58.79 | 45.31 | 41.63 | 93.03 | 76.95 | 84.23 | 36.37 | 72.76 |
| SwinIR | 74.63 | 72.82 | 73.71 | 57.77 | 58.37 | 71.74 | 45.41 | 55.62 | 46.40 | 38.52 | 91.92 | 75.52 | 82.92 | 37.52 | 70.82 |
| StableSR | 69.71 | 74.74 | 72.14 | 51.76 | 56.42 | 74.64 | 46.40 | 57.22 | 42.36 | 40.08 | 89.66 | 77.12 | 82.92 | 38.02 | 70.82 |
| SUPIR | 68.78 | 73.06 | 70.85 | 49.43 | 54.86 | 63.38 | 43.90 | 51.87 | 42.38 | 35.02 | 89.05 | 76.17 | 82.11 | 40.24 | 69.65 |
| TAIR | 50.00 | 41.21 | 55.21 | 39.41 | 29.18 | 60.83 | 34.76 | 43.77 | 41.46 | 28.40 | 87.18 | 57.38 | 69.87 | 35.23 | 52.92 |
| GLYPH-SR (ours) | 69.48 | 77.08 | 73.09 | 47.00 | 57.59 | 68.28 | 46.92 | 55.62 | 38.27 | 38.52 | 90.05 | 80.51 | 85.01 | 39.78 | 73.93 |
| **SVT(x8)** | | | | | | | | | | | | | | | |
| LR | 34.62 | 4.84 | 8.49 | 8.92 | 4.43 | 53.04 | 18.82 | 27.78 | 19.86 | 16.13 | 79.78 | 29.23 | 42.79 | 8.65 | 27.22 |
| HR | 74.73 | 87.86 | 80.76 | 27.69 | 67.73 | 81.18 | 82.97 | 82.06 | 34.80 | 69.58 | 87.68 | 82.14 | 84.82 | 17.77 | 73.65 |
| BSRGAN | 35.75 | 9.18 | 14.61 | 13.02 | 7.88 | 36.54 | 7.99 | 13.12 | 14.68 | 7.02 | 76.77 | 15.34 | 25.56 | 6.25 | 14.66 |
| DiffBIR | 25.00 | 12.54 | 16.70 | 16.56 | 9.11 | 29.23 | 13.58 | 18.55 | 18.51 | 10.22 | 44.16 | 14.93 | 22.32 | 10.21 | 12.56 |
| DiffTSR | 28.03 | 6.29 | 10.28 | 11.36 | 5.42 | 31.51 | 6.46 | 10.72 | 14.98 | 5.62 | 62.50 | 9.09 | 15.87 | 4.60 | 8.62 |
| InvSR | 29.34 | 12.08 | 17.12 | 18.54 | 9.36 | 37.80 | 14.68 | 21.15 | 19.87 | 11.82 | 50.00 | 13.73 | 21.54 | 8.20 | 12.07 |
| MARCONet | 0.00 | 0.00 | 0.00 | 0.00 | 0.00 | 0.00 | 0.00 | 0.00 | 2.30 | 0.00 | 0.00 | 0.00 | 0.00 | 0.05 | 0.00 |
| MARCONet++ | 28.03 | 6.29 | 10.28 | 12.41 | 5.42 | 38.36 | 8.54 | 13.97 | 14.26 | 7.51 | 62.96 | 15.60 | 22.32 | 6.62 | 14.29 |
| PiSA-SR | 36.11 | 11.57 | 17.53 | 14.53 | 9.61 | 47.84 | 16.06 | 24.05 | 19.83 | 13.67 | 79.41 | 24.77 | 37.76 | 7.72 | 23.28 |
| Real-ESRGAN | 34.50 | 11.93 | 17.73 | 16.17 | 9.73 | 48.20 | 15.35 | 23.29 | 19.45 | 13.18 | 76.68 | 19.30 | 30.83 | 7.14 | 18.23 |
| SwinIR | 34.59 | 9.26 | 14.61 | 13.61 | 7.88 | 45.63 | 13.43 | 20.75 | 17.89 | 11.58 | 76.44 | 19.04 | 30.48 | 6.45 | 17.98 |
| StableSR | 41.13 | 14.05 | 20.95 | 17.50 | 11.70 | 50.45 | 16.12 | 24.43 | 19.15 | 13.92 | 79.43 | 29.71 | 43.24 | 9.96 | 27.59 |
| SUPIR | 42.82 | 27.66 | 33.61 | 15.29 | 20.20 | 43.00 | 30.90 | 35.96 | 18.80 | 21.92 | 59.22 | 26.68 | 36.78 | 11.28 | 22.54 |
| TAIR | 35.38 | 15.48 | 21.54 | 16.68 | 12.07 | 40.91 | 16.46 | 23.48 | 21.45 | 13.30 | 51.87 | 17.96 | 26.68 | 10.14 | 15.39 |
| GLYPH-SR (ours) | 48.52 | 49.06 | 48.79 | 19.16 | 32.27 | 57.32 | 55.03 | 56.16 | 23.17 | 39.04 | 69.57 | 50.53 | 58.54 | 17.99 | 41.38 |
| **SCUT-CTW1500(x8)** | | | | | | | | | | | | | | | |
| LR | 27.91 | 0.27 | 0.53 | 1.38 | 0.26 | 38.41 | 2.54 | 4.76 | 8.22 | 2.44 | 73.05 | 5.47 | 10.18 | 5.14 | 5.36 |
| HR | 72.57 | 68.73 | 70.59 | 56.36 | 54.55 | 75.74 | 65.33 | 70.15 | 52.17 | 54.02 | 87.67 | 69.89 | 77.77 | 35.27 | 63.63 |
| BSRGAN | 29.10 | 1.79 | 3.37 | 7.67 | 1.72 | 31.06 | 1.89 | 3.54 | 7.31 | 1.80 | 64.75 | 2.00 | 3.88 | 2.10 | 1.98 |
| DiffBIR | 15.66 | 2.81 | 4.76 | 17.18 | 2.44 | 11.46 | 3.28 | 5.10 | 16.05 | 2.62 | 21.26 | 2.60 | 4.64 | 9.28 | 2.37 |
| DiffTSR | 28.10 | 1.55 | 2.95 | 6.94 | 1.50 | 28.33 | 1.51 | 2.86 | 6.44 | 1.45 | 51.94 | 1.49 | 2.90 | 2.37 | 1.47 |
| InvSR | 21.15 | 1.10 | 2.09 | 7.13 | 1.06 | 18.66 | 1.15 | 2.17 | 7.17 | 1.10 | 55.45 | 1.24 | 2.43 | 1.78 | 1.23 |
| MARCONet | 20.00 | 0.07 | 0.13 | 0.59 | 0.07 | 16.67 | 0.31 | 0.61 | 2.28 | 0.31 | 46.15 | 0.13 | 0.26 | 0.33 | 0.13 |
| MARCONet++ | 35.53 | 1.21 | 2.35 | 4.27 | 1.19 | 34.48 | 1.35 | 2.60 | 5.15 | 1.32 | 67.42 | 1.33 | 2.60 | 1.03 | 1.32 |
| PiSA-SR | 25.50 | 4.48 | 7.61 | 17.41 | 3.96 | 29.21 | 3.92 | 6.92 | 13.26 | 3.58 | 60.45 | 5.20 | 9.43 | 6.82 | 4.95 |
| Real-ESRGAN | 32.77 | 2.72 | 5.02 | 9.82 | 2.57 | 34.74 | 3.07 | 5.64 | 9.91 | 2.90 | 66.30 | 4.11 | 7.74 | 3.81 | 4.02 |
| SwinIR | 36.32 | 1.93 | 3.67 | 6.60 | 1.87 | 32.06 | 2.52 | 4.68 | 9.01 | 2.40 | 67.96 | 2.74 | 5.27 | 2.77 | 2.70 |
| StableSR | 39.90 | 1.74 | 3.33 | 5.63 | 1.69 | 40.55 | 2.34 | 4.43 | 7.41 | 2.26 | 71.37 | 3.95 | 7.49 | 3.68 | 3.89 |
| SUPIR | 19.91 | 3.15 | 5.43 | 14.12 | 2.79 | 22.41 | 3.64 | 6.26 | 13.13 | 3.23 | 33.47 | 3.91 | 7.00 | 7.13 | 3.63 |
| TAIR | 27.45 | 6.67 | 10.74 | 21.66 | 5.67 | 28.06 | 5.52 | 9.23 | 18.45 | 4.84 | 38.99 | 7.19 | 12.14 | 12.97 | 6.46 |
| GLYPH-SR (ours) | 22.61 | 7.35 | 11.09 | 20.54 | 5.87 | 25.24 | 10.38 | 14.71 | 20.10 | 7.94 | 34.12 | 9.34 | 14.67 | 13.85 | 7.92 |
| **CUTE(x8)** | | | | | | | | | | | | | | | |
| LR | 60.00 | 29.30 | 39.38 | 38.33 | 24.51 | 68.33 | 37.44 | 36.31 | 45.45 | 31.91 | 91.02 | 62.81 | 67.18 | 29.99 | 59.14 |
| HR | 77.78 | 80.38 | 79.06 | 51.61 | 65.37 | 75.33 | 51.36 | 61.08 | 45.35 | 43.97 | 95.17 | 79.76 | 86.78 | 37.54 | 76.65 |
| BSRGAN | 58.33 | 52.41 | 55.21 | 47.56 | 38.13 | 68.42 | 35.29 | 46.57 | 42.81 | 30.35 | 91.61 | 58.20 | 71.18 | 28.40 | 55.25 |
| DiffBIR | 61.58 | 57.67 | 59.56 | 45.10 | 42.41 | 58.73 | 36.10 | 44.71 | 38.97 | 28.79 | 88.61 | 58.58 | 70.53 | 30.10 | 54.47 |
| DiffTSR | 60.76 | 49.23 | 54.39 | 48.20 | 37.35 | 62.16 | 32.09 | 42.33 | 41.04 | 26.85 | 86.23 | 50.00 | 63.30 | 27.95 | 46.30 |
| InvSR | 55.80 | 57.06 | 56.42 | 47.28 | 39.30 | 60.98 | 35.89 | 45.18 | 39.50 | 29.18 | 87.43 | 61.86 | 72.46 | 35.41 | 56.81 |
| MARCONet | 100.00 | 1.56 | 3.07 | 1.15 | 1.56 | 33.33 | 2.45 | 4.56 | 4.65 | 2.33 | 100.00 | 1.95 | 3.82 | 0.83 | 1.95 |
| MARCONet++ | 56.49 | 45.79 | 50.58 | 48.26 | 33.85 | 67.57 | 33.94 | 45.18 | 40.53 | 29.18 | 90.51 | 59.09 | 71.50 | 33.08 | 55.64 |
| PiSA-SR | 58.23 | 48.17 | 52.72 | 51.39 | 35.80 | 61.61 | 32.24 | 42.33 | 41.76 | 26.85 | 91.76 | 63.52 | 75.24 | 30.23 | 60.31 |
| Real-ESRGAN | 60.67 | 57.75 | 59.18 | 50.53 | 42.02 | 70.00 | 38.01 | 49.27 | 42.81 | 32.68 | 93.25 | 61.79 | 74.33 | 31.11 | 59.14 |
| SwinIR | 58.33 | 47.40 | 52.30 | 49.71 | 35.41 | 64.66 | 34.72 | 45.18 | 42.14 | 29.18 | 90.68 | 60.33 | 72.46 | 28.55 | 56.81 |
| StableSR | 60.06 | 55.73 | 57.81 | 51.68 | 40.66 | 61.22 | 35.80 | 45.18 | 43.35 | 29.18 | 88.79 | 63.24 | 73.87 | 32.52 | 58.56 |
| SUPIR | 57.69 | 58.33 | 58.01 | 43.46 | 40.86 | 59.83 | 33.33 | 42.81 | 35.98 | 27.24 | 82.25 | 61.23 | 70.20 | 35.16 | 54.09 |
| TAIR | 49.30 | 37.84 | 42.81 | 37.98 | 27.24 | 54.10 | 32.84 | 40.87 | 40.20 | 25.68 | 82.27 | 50.00 | 62.20 | 33.88 | 45.14 |
| GLYPH-SR (ours) | 63.49 | 63.83 | 63.66 | 42.40 | 46.69 | 58.91 | 37.25 | 45.65 | 36.80 | 29.57 | 83.80 | 65.79 | 73.71 | 35.12 | 58.37 |

Table 3: Quantitative comparison of SR models on three scene-text datasets (SVT, SCUT-CTW1500, CUTE80) at ×4 and ×8 upscaling factors. Metrics include distortion-based (PSNR, SSIM, LPIPS↓) and perceptual quality scores (MANIQA, CLIP-IQA, MUSIQ). Red and blue denote the best and second-best results, respectively.

| Dataset | SR model | PSNR | SSIM | LPIPS↓ | MANIQA | CLIP-IQA | MUSIQ |
|---|---|---|---|---|---|---|---|
| SVT(x4) | LR | 30.12 | 87.21 | 33.58 | 20.45 | 17.06 | 26.31 |
| | HR | - | - | - | 23.64 | 25.51 | 41.63 |
| | BSRGAN | 28.09 | 83.16 | 35.34 | 38.16 | 39.63 | 66.25 |
| | DiffBIR | 21.96 | 63.94 | 43.55 | 47.82 | 58.66 | 71.18 |
| | DiffTSR | 26.06 | 78.42 | 44.95 | 21.34 | 27.69 | 46.24 |
| | InvSR | 24.78 | 76.58 | 38.61 | 46.78 | 57.30 | 70.81 |
| | PiSA-SR | 26.58 | 82.04 | 34.13 | 37.41 | 44.30 | 61.87 |
| | MARCONet | 18.40 | 69.07 | 58.91 | 30.92 | 24.43 | 27.26 |
| | MARCONet++ | 27.73 | 84.33 | 34.75 | 29.31 | 19.82 | 49.20 |
| | Real-ESRGAN | 29.67 | 88.58 | 30.68 | 31.16 | 28.58 | 51.14 |
| | SwinIR | 30.48 | 86.38 | 35.29 | 26.32 | 44.50 | 34.55 |
| | StableSR | 30.54 | 87.00 | 33.73 | 24.75 | 32.18 | 24.44 |
| | SUPIR | 22.76 | 67.15 | 45.14 | 42.36 | 48.42 | 67.55 |
| | TAIR | 24.80 | 73.72 | 40.31 | 29.31 | 19.82 | 49.20 |
| | GLYPH-SR (ours) | 22.89 | 67.19 | 42.20 | 47.75 | 59.40 | 70.99 |
| SCUT-CTW1500(x4) | LR | 19.16 | 55.44 | 47.91 | 28.92 | 31.16 | 25.82 |
| | HR | - | - | - | 64.23 | 77.42 | 70.87 |
| | BSRGAN | 20.22 | 64.59 | 32.12 | 51.41 | 47.44 | 67.52 |
| | DiffBIR | 17.91 | 56.34 | 36.20 | 62.37 | 61.90 | 71.19 |
| | DiffTSR | 18.99 | 58.59 | 41.34 | 35.39 | 30.59 | 55.83 |
| | InvSR | 18.32 | 60.71 | 32.99 | 57.75 | 55.94 | 69.25 |
| | PiSA-SR | 20.07 | 63.99 | 31.18 | 56.31 | 53.05 | 68.19 |
| | MARCONet | 14.80 | 40.89 | 66.30 | 33.34 | 16.54 | 28.78 |
| | MARCONet++ | 19.08 | 58.77 | 41.60 | 34.65 | 19.58 | 43.61 |
| | Real-ESRGAN | 20.85 | 67.46 | 36.81 | 40.81 | 43.43 | 52.66 |
| | SwinIR | 19.91 | 58.45 | 47.11 | 33.85 | 46.07 | 39.36 |
| | StableSR | 19.24 | 55.45 | 49.03 | 31.04 | 43.61 | 24.92 |
| | SUPIR | 13.61 | 32.98 | 52.15 | 57.35 | 51.68 | 66.96 |
| | TAIR | 18.19 | 60.04 | 34.46 | 65.38 | 47.05 | 67.08 |
| | GLYPH-SR (ours) | 18.19 | 54.67 | 37.15 | 70.33 | 57.88 | 70.31 |
| CUTE80(x4) | LR | 26.44 | 78.35 | 36.69 | 28.93 | 36.80 | 37.64 |
| | HR | - | - | - | 40.12 | 55.71 | 60.81 |
| | BSRGAN | 27.35 | 79.76 | 31.83 | 44.22 | 55.73 | 69.13 |
| | DiffBIR | 22.60 | 66.07 | 37.74 | 51.04 | 72.64 | 69.06 |
| | DiffTSR | 24.06 | 72.66 | 42.74 | 33.94 | 38.47 | 58.74 |
| | InvSR | 24.41 | 75.55 | 32.93 | 50.30 | 67.78 | 70.66 |
| | PiSA-SR | 25.83 | 77.41 | 31.49 | 45.82 | 61.81 | 66.18 |
| | MARCONet | 16.17 | 63.52 | 56.18 | 33.58 | 26.69 | 31.06 |
| | MARCONet++ | 25.16 | 77.75 | 34.91 | 31.88 | 34.90 | 54.15 |
| | Real-ESRGAN | 28.14 | 82.30 | 32.01 | 38.20 | 48.71 | 60.65 |
| | SwinIR | 27.18 | 77.95 | 38.02 | 31.87 | 59.32 | 47.94 |
| | StableSR | 26.23 | 79.51 | 30.45 | 36.26 | 49.74 | 60.09 |
| | SUPIR | 22.42 | 66.20 | 39.33 | 47.50 | 62.62 | 68.26 |
| | TAIR | 20.82 | 69.03 | 41.27 | 58.25 | 49.76 | 72.06 |
| | GLYPH-SR (ours) | 23.03 | 69.54 | 37.03 | 49.77 | 65.93 | 69.96 |
| SVT(x8) | LR | 26.15 | 78.90 | 48.54 | 19.81 | 44.07 | 22.96 |
| | HR | - | - | - | 23.64 | 25.51 | 41.63 |
| | BSRGAN | 25.13 | 73.71 | 45.64 | 37.14 | 37.58 | 62.83 |
| | DiffBIR | 22.89 | 65.20 | 50.07 | 45.54 | 53.20 | 64.11 |
| | DiffTSR | 24.45 | 76.19 | 46.32 | 21.39 | 26.39 | 43.96 |
| | InvSR | 22.82 | 71.34 | 41.84 | 32.51 | 50.83 | 51.69 |
| | PiSA-SR | 26.12 | 77.64 | 50.83 | 34.02 | 18.39 | 30.24 |
| | MARCONet | 18.68 | 69.49 | 58.71 | 30.84 | 24.76 | 27.02 |
| | MARCONet++ | 25.15 | 76.11 | 44.44 | 20.97 | 8.44 | 38.06 |
| | Real-ESRGAN | 25.69 | 80.28 | 41.92 | 28.38 | 17.86 | 43.01 |
| | SwinIR | 26.48 | 78.16 | 48.01 | 22.05 | 26.68 | 30.33 |
| | StableSR | 26.38 | 78.15 | 50.20 | 23.16 | 23.38 | 16.22 |
| | SUPIR | 21.23 | 59.08 | 51.46 | 40.17 | 45.06 | 65.20 |
| | TAIR | 22.72 | 68.24 | 44.86 | 31.99 | 29.49 | 54.27 |
| | GLYPH-SR (ours) | 21.77 | 61.36 | 47.85 | 47.40 | 56.78 | 69.93 |
| SCUT-CTW1500(x8) | LR | 17.04 | 43.70 | 63.73 | 16.39 | 26.19 | 17.71 |
| | HR | - | - | - | 64.23 | 77.42 | 70.87 |
| | BSRGAN | 17.32 | 48.50 | 47.86 | 46.21 | 37.83 | 66.05 |
| | DiffBIR | 15.78 | 43.47 | 50.05 | 54.75 | 49.89 | 63.16 |
| | DiffTSR | 14.83 | 40.25 | 54.50 | 35.49 | 31.88 | 50.43 |
| | InvSR | 11.81 | 30.68 | 65.88 | 29.65 | 29.62 | 40.29 |
| | PiSA-SR | 17.22 | 47.63 | 48.90 | 41.77 | 36.75 | 58.95 |
| | MARCONet | 14.93 | 40.72 | 66.71 | 33.56 | 16.20 | 28.95 |
| | MARCONet++ | 16.79 | 44.04 | 57.88 | 14.75 | 8.06 | 30.27 |
| | Real-ESRGAN | 17.65 | 52.34 | 52.14 | 28.37 | 20.95 | 39.99 |
| | SwinIR | 17.39 | 45.42 | 61.26 | 19.00 | 24.14 | 25.64 |
| | StableSR | 17.00 | 43.50 | 66.02 | 20.93 | 20.92 | 16.62 |
| | SUPIR | 12.63 | 26.51 | 58.63 | 55.46 | 47.02 | 65.55 |
| | TAIR | 16.02 | 45.70 | 45.73 | 63.60 | 36.57 | 66.38 |
| | GLYPH-SR (ours) | 16.27 | 41.31 | 52.58 | 61.94 | 48.21 | 63.43 |
| CUTE80(x8) | LR | 23.25 | 71.52 | 47.61 | 17.29 | 22.58 | 17.32 |
| | HR | - | - | - | 40.12 | 55.71 | 60.81 |
| | BSRGAN | 23.84 | 72.55 | 39.14 | 42.07 | 54.31 | 67.33 |
| | DiffBIR | 22.77 | 65.36 | 41.79 | 47.53 | 62.09 | 64.62 |
| | DiffTSR | 22.67 | 70.41 | 42.80 | 33.55 | 42.95 | 57.47 |
| | InvSR | 21.83 | 70.76 | 38.05 | 37.66 | 62.43 | 57.69 |
| | PiSA-SR | 23.36 | 70.52 | 47.71 | 30.71 | 30.80 | 45.16 |
| | MARCONet | 16.37 | 63.64 | 56.08 | 33.66 | 26.69 | 30.88 |
| | MARCONet++ | 22.79 | 70.85 | 43.34 | 21.03 | 22.72 | 44.24 |
| | Real-ESRGAN | 24.01 | 75.58 | 39.60 | 35.17 | 36.46 | 56.55 |
| | SwinIR | 23.82 | 71.22 | 47.71 | 22.72 | 40.40 | 39.44 |
| | StableSR | 7.94 | 35.66 | 79.75 | 26.00 | 40.42 | 34.48 |
| | SUPIR | 20.64 | 61.31 | 43.76 | 46.38 | 61.67 | 67.04 |
| | TAIR | 20.49 | 67.69 | 42.35 | 37.11 | 36.84 | 55.06 |
| | GLYPH-SR (ours) | 21.19 | 65.15 | 42.31 | 47.75 | 65.85 | 68.85 |

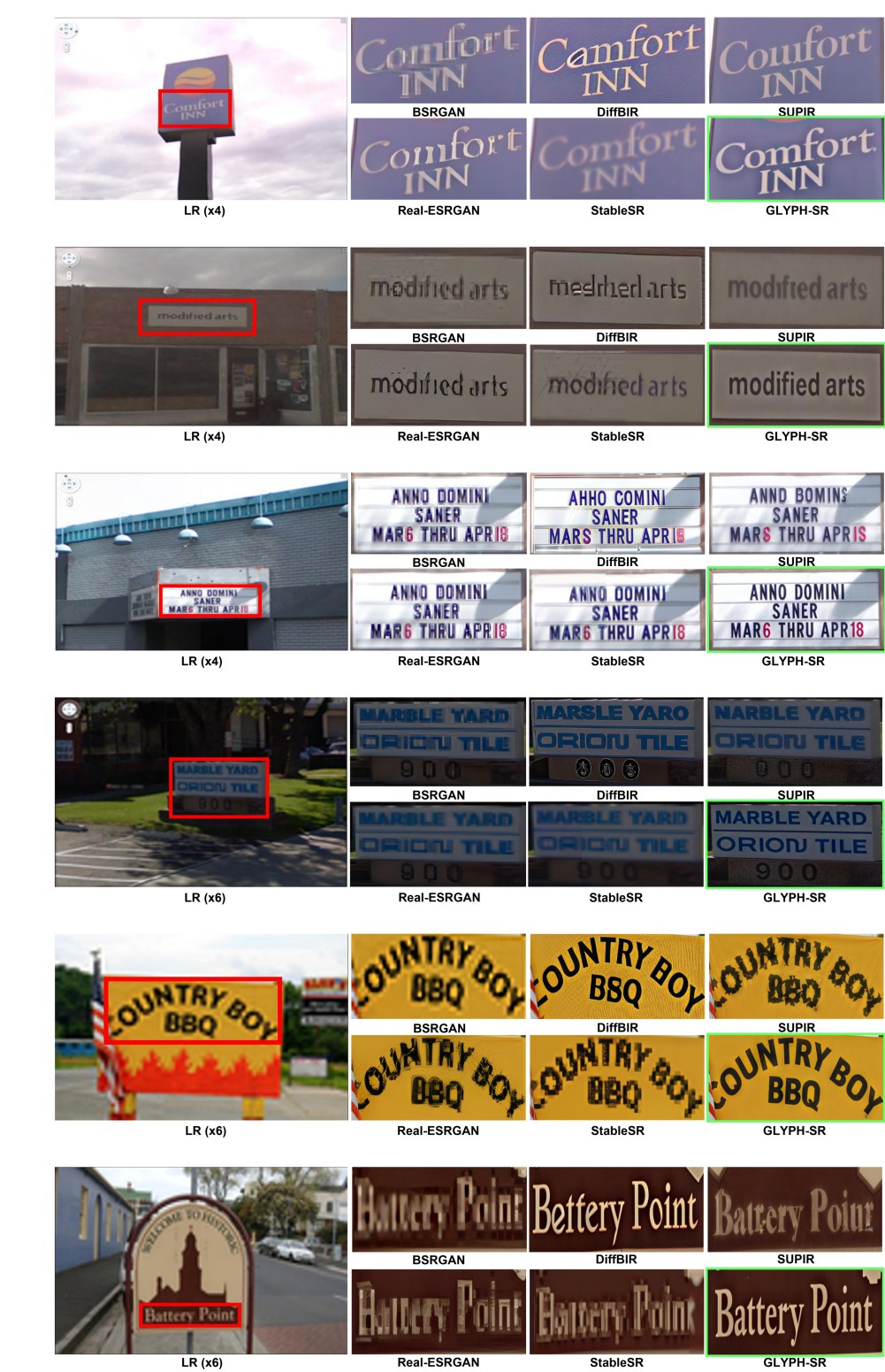

Figure 14: Qualitative comparison.

two distinct conditions: a baseline with accurate text guidance versus a perturbed setting with high guidance error (OCR error). As observed in the heatmaps, the introduction of incorrect conditioning

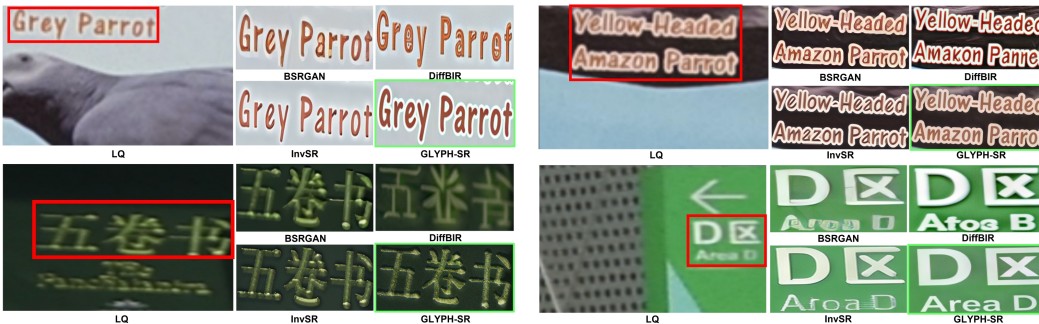

Figure 15: Qualitative comparison of text-focused SR on the Real-Text dataset. Each block shows a LQ input (left) and the outputs of four representative SR methods: BSRGAN, DiffBIR, InvSR, and the proposed GLYPH-SR (right). GLYPH-SR consistently reconstructs sharper glyph boundaries, preserves accurate character shapes, and restores correct textual semantics across diverse scenes—including license plates, bird labels, warning signs, Chinese characters, and directional symbols—while competing methods often exhibit distortions, blurring, or hallucinated characters.

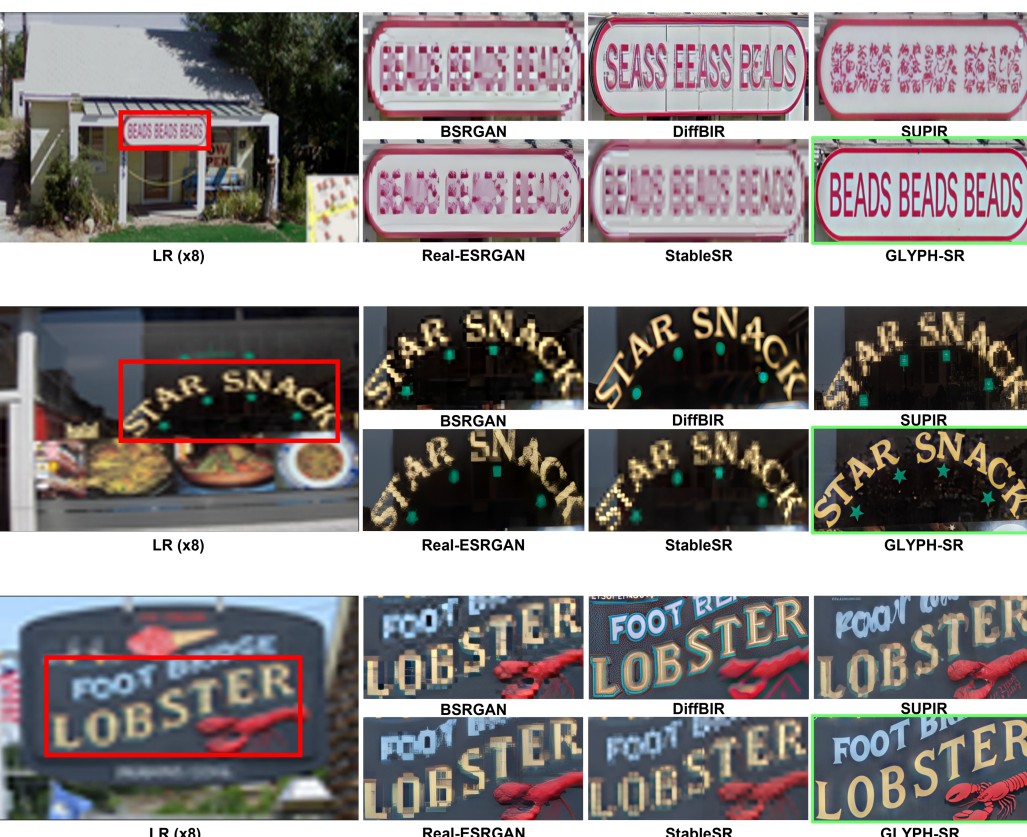

Figure 16: Qualitative comparison of scene-text SR under various degradation scales (×4, ×6, ×8). While prior methods often blur or hallucinate characters, **GLYPH-SR** accurately restores readable, coherent text. Zoom in for detail.

generates significant spatial anomalies, which we term 'prediction noise outliers' (highlighted in red boxes). Unlike the baseline, where noise predictions converge coherently, the high-error setting exhibits persistent instability throughout the intermediate steps (e.g., Step 25).

Based on these visual findings, we identify the mechanism by which incorrect text conditioning degrades the entire image, including non-textual regions:

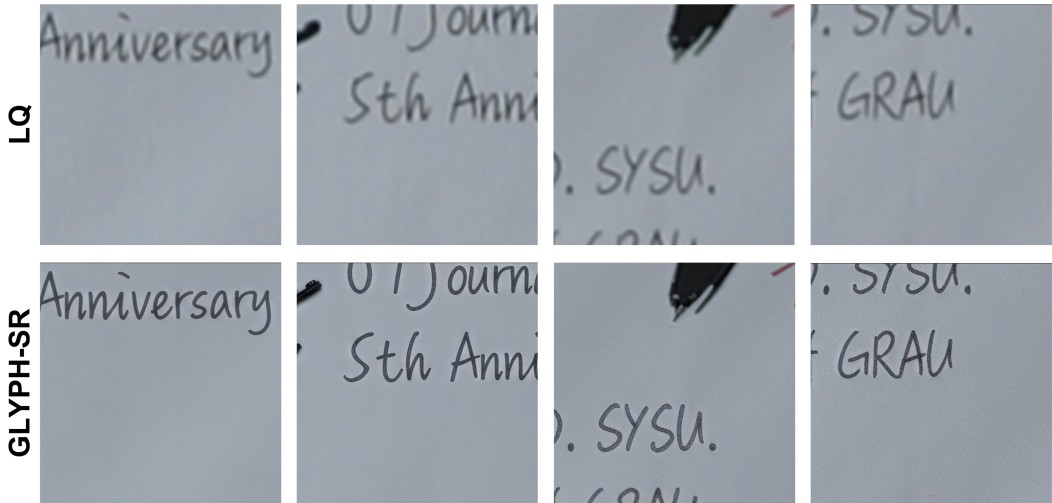

Figure 17: Qualitative Comparison of Handwriting SR. Visual comparison between the LQ input (top row) and the output generated by GLYPH-SR (bottom row) on various handwritten texts.

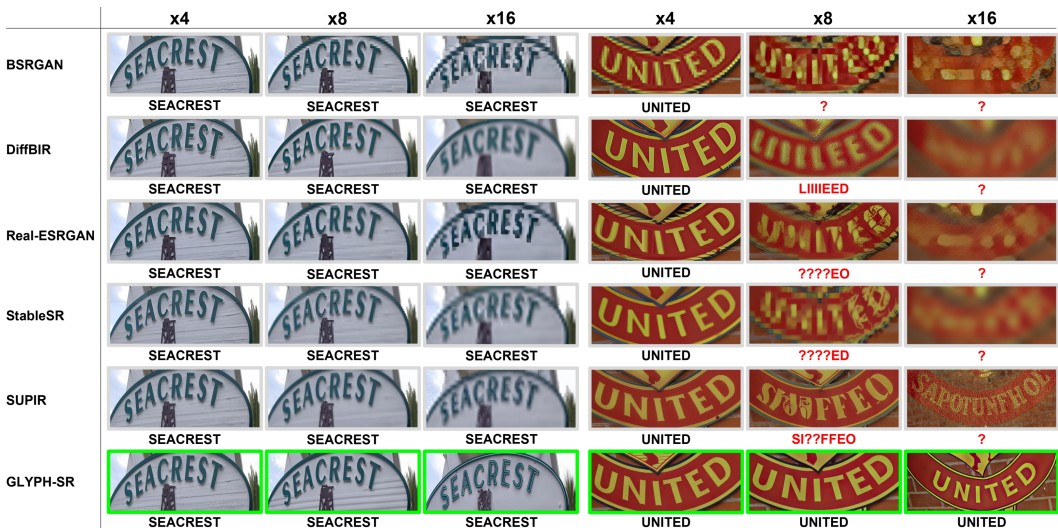

Figure 18: Qualitative comparison of text-centric SR results at ×4, ×8 and ×16 scales.

Table 4: Ablation on the scheduler policy evaluated on the CUTE80 dataset.

(a) CUTE80 (LR × 4)

| Scheduler Policy | MANIQA | CLIP-IQA | MUSIQ | OCR $F_1$ |
|---|---|---|---|---|
| Binary ping–pong | 49.77 | 65.93 | 69.96 | **85.01** |
| Mixing ($\lambda_t = 0.1$) | **49.95** | **70.64** | **70.67** | 81.57 |
| Mixing ($\lambda_t = 0.3$) | 49.04 | 69.56 | 69.75 | 83.18 |
| Mixing ($\lambda_t = 0.5$) | 47.57 | 65.47 | 68.95 | 84.23 |
| Mixing ($\lambda_t = 0.7$) | 47.86 | 68.91 | 68.83 | 81.84 |
| Mixing ($\lambda_t = 0.9$) | 48.85 | 69.11 | 69.13 | 82.65 |

(b) CUTE80 (LR × 8)

| Scheduler Policy | MANIQA | CLIP-IQA | MUSIQ | OCR $F_1$ |
|---|---|---|---|---|
| Binary ping–pong | 47.75 | 65.85 | 68.85 | **73.71** |
| Mixing ($\lambda_t = 0.1$) | **48.89** | 67.65 | **69.56** | 66.49 |
| Mixing ($\lambda_t = 0.3$) | 47.44 | **68.31** | 68.86 | 69.87 |
| Mixing ($\lambda_t = 0.5$) | 46.57 | 64.07 | 67.35 | 73.40 |
| Mixing ($\lambda_t = 0.7$) | 45.80 | 67.98 | 67.19 | 66.84 |
| Mixing ($\lambda_t = 0.9$) | 45.58 | 67.66 | 67.18 | 68.88 |

- **Noise Dispersion:** When incorrect conditioning is injected, the predicted noise values fail to localize around the text regions. Instead, as shown in Fig. 22, high-variance noise artifacts become spatially dispersed across the entire domain.

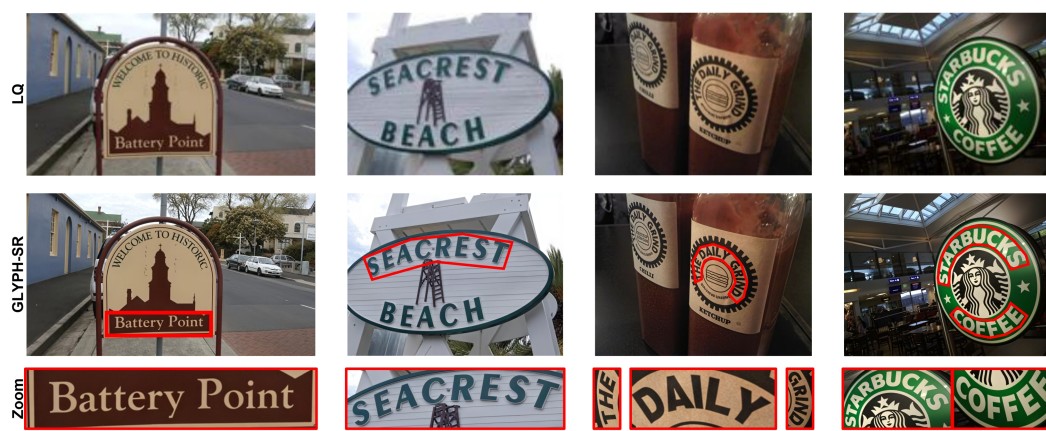

Figure 19: Qualitative Comparison of Curved Text SR (Curved Stress Test). The figure compares the low-quality (LQ) input (top row) with the output of GLYPH-SR (middle row) and a magnified view (bottom row). The curvature of the text increases from left to right across the figure. GLYPH-SR successfully reconstructs the intricate curved glyphs (e.g., 'SEACREST BEACH' and 'STARBUCKS COFFEE') without introducing the shape distortion or blurring often seen in other SR methods. The results demonstrate the robustness of GLYPH-SR in handling non-linear text layouts.

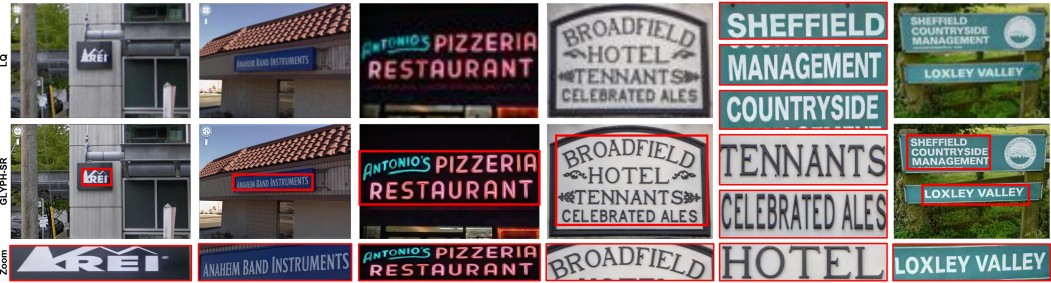

Figure 20: Qualitative Comparison of High-Density Text SR (Dense Stress Test). This comparison highlights the capability of GLYPH-SR (middle row) versus the LQ input (top row) in highly dense and cluttered text environments. The density and complexity of the text increase from left to right across the figure. GLYPH-SR excels at isolating and sharpening individual characters and lines of text, even when tightly packed (e.g., the 'BROADFIELD HOTEL' sign and the multi-line management sign), proving its superior performance in complex, dense scenes where competing methods often struggle with inter-line blurring.

Table 5: Ablation study on CFG scale $w$. Evaluated on CUTE80 ($\times 4$).

| CFG Scale ($w$) | OCR Metric ($F_1$ Score $\uparrow$) | | | SR Metric ($\uparrow$) | | |
|---|---|---|---|---|---|---|
| | OpenOCR | GOT-OCR | LLaVA-NeXT | MANIQA | CLIP-IQA | MUSIQ |
| 2 | **75.54** | **57.62** | 87.53 | 41.26 | 55.96 | 65.23 |
| 4 | 74.63 | 57.22 | **88.50** | 43.42 | 60.01 | 67.30 |
| 7.5* | 73.09 | 55.62 | 85.01 | 49.77 | 65.93 | 69.96 |
| 12 | 71.50 | 53.56 | 84.23 | 49.52 | 69.87 | 70.51 |
| 20 | 65.45 | 45.18 | 79.06 | **50.60** | **74.30** | **71.03** |

- **Dissipation of Predictive Capacity:** In diffusion models, the noise predictor $\epsilon_\theta(z_t, t, c)$ estimates a noise vector subject to a magnitude constraint at each timestep. When the semantic condition $c$ conflicts with the visual features (due to erroneous OCR), the model expends its finite predictive capacity attempting to resolve this ambiguity. Consequently, the prediction energy is dissipated globally rather than being concentrated on glyph restoration.

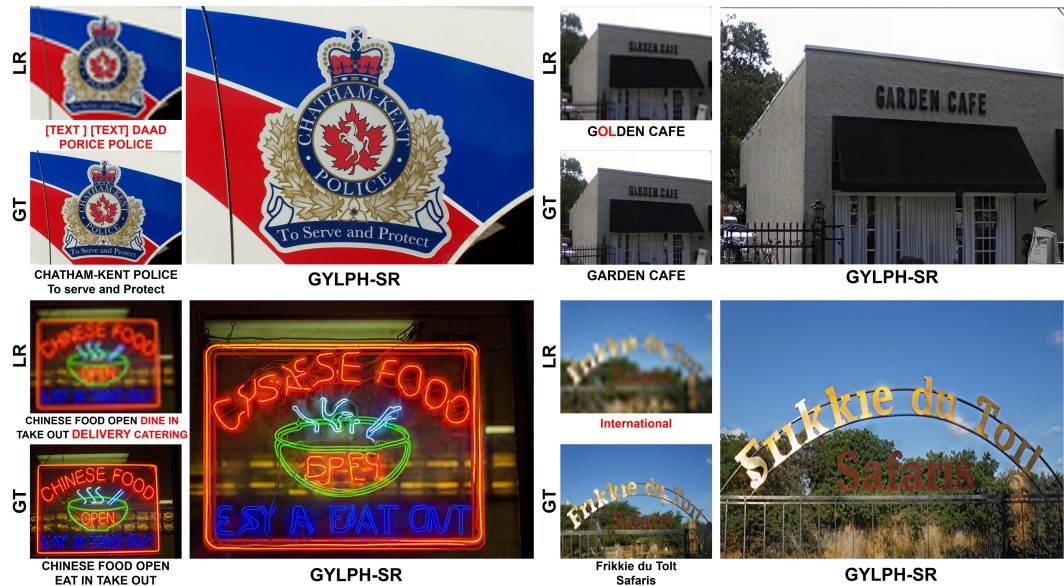

Figure 21: **Performance Comparison under Extreme Degradation.** We compare SR results with extremely LR input images. Despite the inherent difficulty of achieving full restoration in these Extreme LR scenarios, GLYPH-SR demonstrates a unique capability for semantic reconstruction. As shown by the comparison against the LR input and the GT, GLYPH-SR successfully recovers partial text structure and generates sharper outlines, showcasing its effectiveness even when input quality is severely degraded.

Table 6: Ablation study on Control Scale $s_{CTRL}$. Evaluated on CUTE80 ($\times 4$).

| $s_{CTRL}$ | **OCR Metric ($F_1$ Score $\uparrow$)** | | | **SR Metric ($\uparrow$)** | | |
|---|---|---|---|---|---|---|
| | **OpenOCR** | **GOT-OCR** | **LLaVA-NeXT** | **MANIQA** | **CLIP-IQA** | **MUSIQ** |
| **1.0\*** | **73.09** | **55.62** | **85.01** | **49.77** | 65.93 | **69.96** |
| 2.0 | 69.21 | 56.02 | 81.84 | 26.36 | **66.95** | 49.92 |
| 3.0 | 4.56 | 6.04 | 25.17 | 37.17 | 45.28 | 49.21 |
| 10.0 | 0.00 | 0.00 | 0.00 | 35.22 | 25.26 | 27.18 |

Table 7: **Sensitivity to OCR/VLM guidance errors.** Values in parentheses are absolute deltas from the baseline (lower is worse).

| Error rate / Type | OpenOCR $F_1$ | GOT-OCR $F_1$ | MANIQA | CLIP-IQA |
|---|---|---|---|---|
| Baseline | 48.82 | 38.36 | 62.01 | 79.69 |
| 30% | 38.36 ($-10.46$) | 28.67 ($-9.69$) | 45.87 ($-16.14$) | 63.65 ($-16.04$) |
| 50% | 32.03 ($-16.79$) | 26.35 ($-12.01$) | 45.39 ($-16.62$) | 64.88 ($-14.81$) |
| 90% | 27.52 ($-21.30$) | 26.35 ($-12.01$) | 45.61 ($-16.40$) | 66.00 ($-13.59$) |
| Swap | 39.88 ($-8.94$) | 33.12 ($-5.24$) | 45.81 ($-16.20$) | 66.00 ($-13.69$) |
| Drop | 41.85 ($-6.97$) | 32.03 ($-6.33$) | 44.82 ($-17.19$) | 65.30 ($-14.39$) |

- **Global Texture Degradation:** This misallocation of predictive resources leaves insufficient residual capacity for restoring fine-grained non-text textures. As a result, the background is not merely neglected; it is actively degraded by incoherent noise updates driven by the semantic conflict.

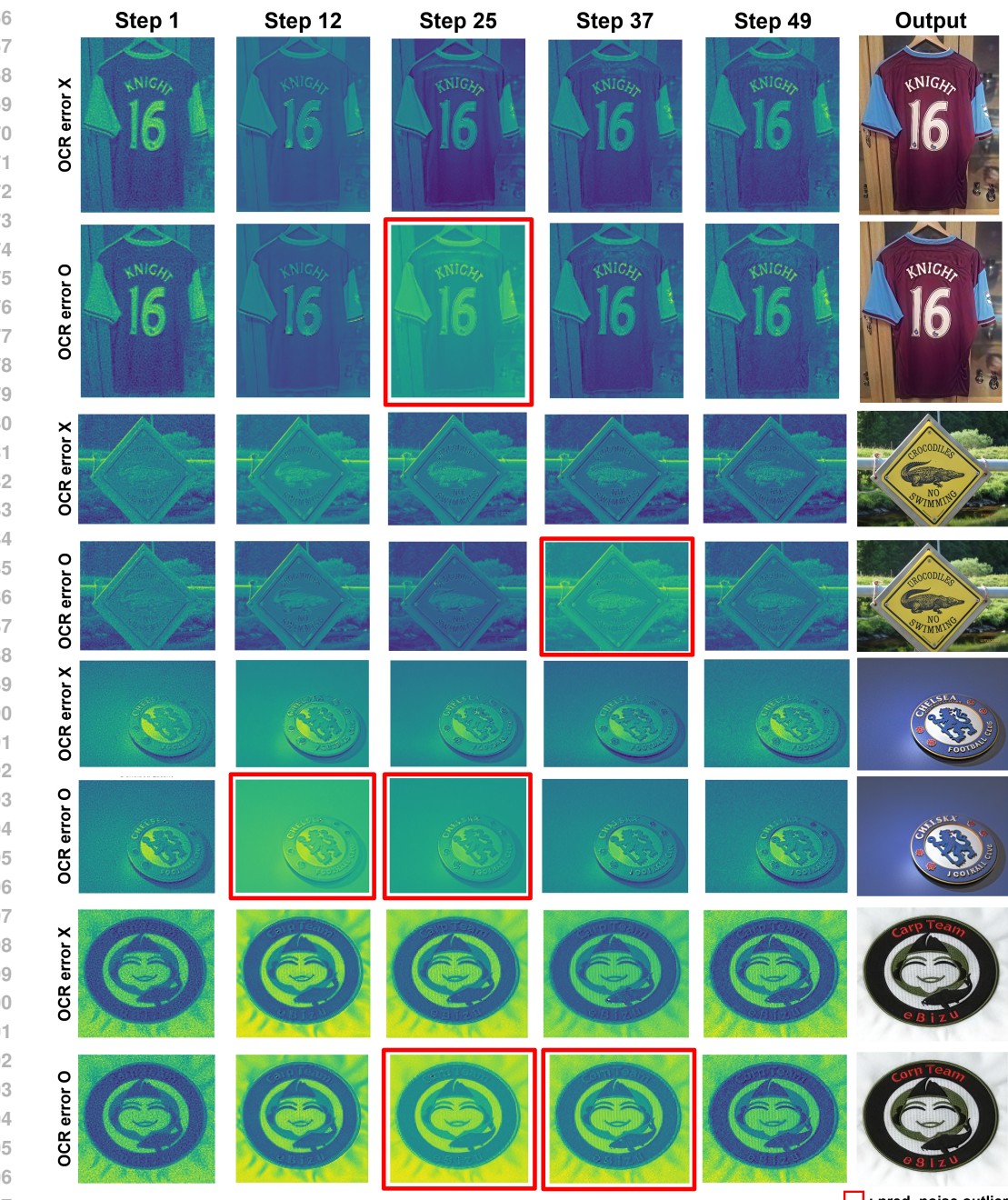

Figure 22: Intermediate Feature Analysis under OCR Error Rates.

Consequently, we posit that text fidelity and background quality are positively correlated in our framework. Precise text guidance acts as a stabilizing anchor, preventing noise dispersion and ensuring a coherent denoising trajectory for both glyphs and scene textures.

As illustrated in Fig. 23, GLYPH-SR can deliver visually plausible SR results yet still *hallucinates* glyphs in regions that were originally non-textual. This deficiency in text-region localization means the reconstructed text may be ambiguous, incomplete, or entirely spurious. Furthermore, when multiple words are present, the model tends to enhance only the most visually salient word and overlook the rest. These failure cases underline the necessity for finer-grained attention mechanisms and explicit supervision of glyph positions in future work.

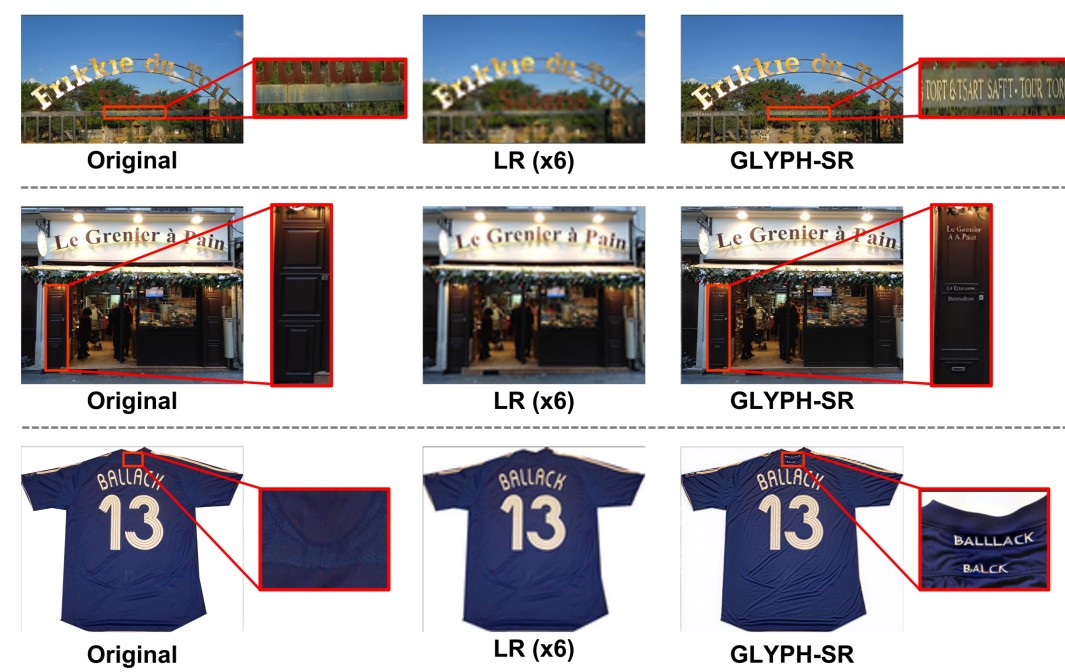

Figure 23: Failure cases where GLYPH-SR produces visually plausible SR outputs but incorrectly generates text in non-textual regions.

## C.4    COMPUTATIONAL FOOTPRINT AND PRACTICAL EFFICIENCY

**Setup.** We benchmark inference on a $4\times$ SR task with $512\times512$ inputs. Times are mean $\pm$ std. over repeated runs. For methods that require a large VLM (SUPIR and GLYPH-SR), we used two NVIDIA A6000 GPUs; reported peak VRAM is the sum across both devices.

Table 8: **Compute comparison.** For VLM-guided methods, #Params lists (*restoration*, *VLM*) in millions.

| Method | #Params (M) | Inference (s / sample) | Peak VRAM (GB) |
|---|---|---|---|
| StableSR | 153 | $79.98 \pm 0.22$ | 10.10 |
| DiffBIR | 385 | $53.14 \pm 1.41$ | 9.64 |
| SUPIR | 18, 152 | $25.25 \pm 0.86$ | 46.21 |
| GLYPH-SR | 13, 225 | $38.25 \pm 1.28$ | 43.56 |

GLYPH-SR trades extra parameters and memory for markedly better text fidelity: it couples a restoration backbone with a powerful OCR/VLM to reason about low-resolution text. This design improves accuracy but introduces a computational bottleneck. To mitigate the cost while keeping readability gains, we will pursue:

- **Lighter VLM Guidance.** Replace the current general-purpose VLM with a compact, LR-text-specialized guider (or distill the guider), reducing parameter count and latency with minimal loss in guidance quality.

- **Inference Optimization ("Block Caching").** Cache and reuse guidance features that repeat across diffusion steps/tiles (e.g., projected text embeddings and cross-attention KV maps), skipping redundant compute and lowering end-to-end runtime.

These directions aim to preserve GLYPH-SR's strengths ("looks right and reads right") while improving deployability under realistic compute budgets.

| Dataset / Guider | OCR metric: $F_1 \uparrow$ | | | SR metric | | |
|---|---|---|---|---|---|---|
| | OpenOCR | GOT-OCR | LLaVA-NeXT | MANIQA | CLIP-IQA | MUSIQ |
| Ours | 73.09 | 55.62 | 85.01 | 49.77 | 65.93 | 69.96 |
| R-4B | 72.77 | 55.21 | 84.75 | 40.78 | 57.20 | 65.57 |
| PaliGemma-3B | 70.53 | 51.45 | 84.52 | 42.11 | 59.61 | 65.83 |

Table 9: Compact/alternative guiders in GLYPH-SR.

Table 9 highlights a clear but controllable accuracy–efficiency trade-off. While GLYPH-SR employs a high-capacity LLaVA-NeXT-8B guider by default to handle severe degradations, our ablation study demonstrates that significantly lighter models can serve as efficient alternatives. Specifically, replacing the 8B guider with the 4B model (R-4B) reduces the model size by 50%, yet the OpenOCR $F_1$ score drops by a negligible 0.32 points (from 73.09 to 72.77). Even with the 3B model (PaliGemma-3B), which represents a ∼62.5% reduction in parameters, the model maintains robust text recognition capabilities with a moderate decline of 2.56 points in OpenOCR. This quantitative evidence indicates that practitioners with tighter deployment budgets can adopt these lighter guiders to substantially reduce computational overhead (latency and VRAM) at only a modest performance cost, proving GLYPH-SR's adaptability to resource-constrained environments.

Table 10: Preliminary evaluation on CUTE80 (×16) comparing the baseline GLYPH-SR with proposed solution strategies. Red indicates improvement, and Blue indicates degradation compared to the baseline.

| Method | OpenOCR | GOT-OCR | ManiQA | ClipIQA |
|---|---|---|---|---|
| LR | 12.41 | 12.41 | 14.12 | 29.11 |
| GLYPH-SR (Baseline) | 9.63 | 11.72 | 38.52 | 54.68 |
| GLYPH-SR + [1] | **13.77** (+4.14) | 12.41 (+0.69) | 35.75 (-2.77) | 49.68 (-5.00) |
| GLYPH-SR + CoT | 10.33 (+0.70) | **13.09** (+1.37) | **42.15** (+3.63) | **59.99** (+5.31) |

We conducted a proof-of-concept experiment on the CUTE80 dataset under an extreme downscaling factor of ×16, a setting where severe degradation renders text nearly illegible and prone to detection failures. As summarized in Table 10, integrating the dual-stage restoration framework [26] significantly improved OpenOCR performance (+4.14), demonstrating its effectiveness in mitigating hallucinations and recovering structural details despite detection errors. Furthermore, refining the inference process with Chain-of-Thought (CoT) prompting notably enhanced perceptual quality metrics (ManiQA and ClipIQA) while boosting recognition accuracy, confirming its capacity to effectively balance the trade-off between fidelity and perception. These results empirically verify the robust extendability of GLYPH-SR even under extreme degradation scenarios.

**Trainable parameters.** Although the full model size is large due to the VLM, our *fine-tuning* recipe is lightweight. We freeze the diffusion backbone and update only two components:

1. **TS-ControlNet branch** (≈54.8M parameters) that handles text-guidance fusion.

2. **VLM LoRA adapter** (≈5.9M parameters) with low rank ($r$=8), `lora_alpha` of 32, and dropout of 0.05.

To minimize memory further, the large *frozen* VLM is loaded in 4-bit quantization (`nf4` with double quantization via BitsAndBytes).

Table 11: **Trainable parameter counts** (millions). Despite using a VLM, GLYPH-SR keeps *trainable* parameters modest via freezing and LoRA.

| Metric | GLYPH-SR | PiSA-SR | StableSR | DiffBIR | SUPIR | DiffTSR |
|---|---|---|---|---|---|---|
| Trainable (M) | **60.7** | 0.38 | 152.67 | 378.95 | 3865.64 | 55.31 |

**Inference latency.** The OCR/VLM guider is the main overhead driver. Out of a total per-image latency of $38.25 \pm 1.28$ seconds (Sec. C.4), the VLM component accounts for $\approx 8.46$ seconds. Notably, while integrating the VLM increases total parameter count, the latency impact is not proportional. In practice, we retain training practicality with only 60.7M trainable parameters and observe that the rise in inference time is moderate relative to the parameter growth, yielding a favorable trade-off between accuracy (readability and IQA) and compute.

**Implication.** These results align with our compute study (Table 8): GLYPH-SR deliberately expends parameters on guidance quality to secure text fidelity, yet its fine-tuning footprint remains compact and deployable. Further efficiency gains are compatible with our design (e.g., lighter LR-text guiders and block caching for reusable guidance features).

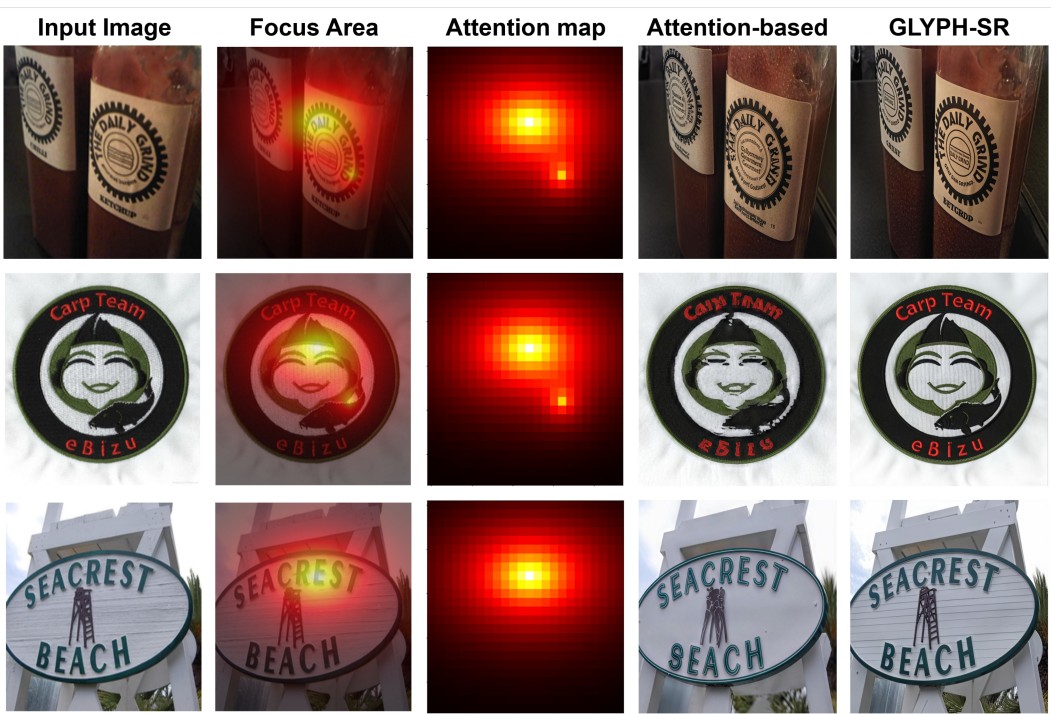

Figure 24: Comparison with explicit attention injection. We investigated an alternative design that explicitly injects OCR attention maps into the backbone. However, this explicit attention guidance proves detrimental; instead of refining features, it interferes with the backbone's pre-trained priors, causing the model to focus on irrelevant noise (as shown in the Attention Map). In contrast, our final GLYPH-SR (rightmost) harmonizes text guidance without such interference, demonstrating successful restoration.

## REFERENCES FOR APPENDIX

[12] Sidi Yang, Tianhe Wu, Shuwei Shi, Shanshan Lao, Yuan Gong, Mingdeng Cao, Jiahao Wang, and Yujiu Yang. Maniqa: Multi-dimension attention network for no-reference image quality assessment. In Proceedings of the IEEE/CVF Conference on Computer Vision and Pattern Recognition Workshops (CVPRW), pages 2286–2295, 2022.

[13] Jianyi Wang, Kelvin CK Chan, and Chen Change Loy. Exploring clip for assessing the look and feel of images. In AAAI, 2023.

[14] Jun Ke, Guy Hacohen, Phillip Isola, William T. Freeman, Michael Rubinstein, and Eli Shechtman. Musiq: Multi-scale image quality assessment. In Proceedings of the IEEE/CVF International Conference on Computer Vision (ICCV), pages 8827–8837, 2021.

[18] Haotian Liu, Chunyuan Li, Yuheng Li, Bo Li, Yuanhan Zhang, Sheng Shen, and Yong Jae Lee. Llava-next: Improved reasoning, ocr, and world knowledge, January 2024.

[20] Fanghua Yu, Jinjin Gu, Zheyuan Li, Jinfan Hu, Xiangtao Kong, Xintao Wang, Jingwen He, Yu Qiao, and Chao Dong. Scaling up to excellence: Practicing model scaling for photo-realistic image restoration in the wild. In Proceedings of the IEEE/CVF Conference on Computer Vision and Pattern Recognition, pages 25669–25680, 2024.

[21] Richard Zhang, Phillip Isola, Alexei A. Efros, Eli Shechtman, and Oliver Wang. The unreasonable effectiveness of deep features as a perceptual metric. In Proceedings of the IEEE Conference on Computer Vision and Pattern Recognition (CVPR), June 2018.

[22] Yochai Blau and Tomer Michaeli. The perception-distortion tradeoff. In Proceedings of the IEEE Conference on Computer Vision and Pattern Recognition (CVPR), June 2018.

[23] Gu Jinjin, Cai Haoming, Chen Haoyu, Ye Xiaoxing, Jimmy S Ren, and Dong Chao. Pipal: a large-scale image quality assessment dataset for perceptual image restoration. In Computer Vision–ECCV 2020: 16th European Conference, Glasgow, UK, August 23–28, 2020, Proceedings, Part XI 16, pages 633–651. Springer, 2020.

[24] Jinjin Gu, Haoming Cai, Chao Dong, Jimmy S. Ren, Radu Timofte, Yuan Gong, Shanshan Lao, Shuwei Shi, Jiahao Wang, Sidi Yang, Tianhe Wu, Weihao Xia, Yujiu Yang, Mingdeng Cao, Cong Heng, Lingzhi Fu, Rongyu Zhang, Yusheng Zhang, Hao Wang, Hongjian Song, Jing Wang, Haotian Fan, Xiaoxia Hou, Ming Sun, Mading Li, Kai Zhao, Kun Yuan, Zishang Kong, Mingda Wu, Chuanchuan Zheng, Marcos V. Conde, Maxime Burchi, Longtao Feng, Tao Zhang, Yang Li, Jingwen Xu, Haiqiang Wang, Yiting Liao, Junlin Li, Kele Xu, Tao Sun, Yunsheng Xiong, Abhisek Keshari, Komal, Sadbhawana Thakur, Vinit Jakhetiya, Badri N Subudhi, Hao-Hsiang Yang, Hua-En Chang, Zhi-Kai Huang, Wei-Ting Chen, Sy-Yen Kuo, Saikat Dutta, Sourya Dipta Das, Nisarg A. Shah, and Anil Kumar Tiwari. Ntire 2022 challenge on perceptual image quality assessment. In Proceedings of the IEEE/CVF Conference on Computer Vision and Pattern Recognition (CVPR) Workshops, pages 951–967, June 2022.

[25] Fanghua Yu, Jinjin Gu, Zheyuan Li, Jinfan Hu, Xiangtao Kong, Xintao Wang, Jingwen He, Yu Qiao, and Chao Dong. Scaling up to excellence: Practicing model scaling for photo-realistic image restoration in the wild. In Proceedings of the IEEE/CVF Conference on Computer Vision and Pattern Recognition (CVPR), pages 25669–25680, June 2024.

[26] Sung, Mingyu, et al. "A Novel VLM-Guided Diffusion Model for Remote Sensing Image Super-Resolution." IEEE Geoscience and Remote Sensing Letters 22 (2025): LGRS-2025.