# OpenReview forum: "GLYPH-SR: Can We Achieve Both High-Quality Image Super-Resolution and High-Fidelity Text Recovery via VLM-Guided Latent Diffusion Model?"
_ICLR.cc/2026/Conference — ICLR 2026 Conference Withdrawn Submission_

### Official Review · Reviewer_DpJe · 2025-10-18

**Soundness:** 3
**Presentation:** 4
**Contribution:** 2
**Rating:** 4
**Confidence:** 4

**Summary:**

The paper treats scene-text SR as a bi-objective problem: visual quality + text legibility. It uses a dual-branch Text-SR Fusion ControlNet guided by OCR text/positions and a ping-pong scheduler that alternates text-centric and image-centric guidance. It reports big gains in OCR F1 on SVT/CTW/CUTE80 while keeping perceptual metrics competitive.

**Strengths:**

1. Clear goal (make text readable, not just “look sharp”).
2. Comprehensive evaluation with OCR metrics and perceptual IQA.

**Weaknesses:**

1. Limited analysis of trade-offs (e.g., when text gets clearer, what happens to non-text textures?).
2. No multilingual or curved-text stress test.
3. Sensitivity to OCR detector quality is not studied.

**Questions:**

1. How robust is the method to OCR detection errors?
2. Can the approach handle dense, multi-language street scenes?
3. How is the ping-pong schedule chosen; can it be learned?

---

> ### Author Response · Authors · 2025-11-24
>
> # [Response to DpJe. Weakness 1]
>
> To address the Reviewer's concern, in the revised version, we have strengthened the analysis on trade-offs and clarified what happens to non-text textures when text gets clearer. Specifically, we have made the following corrections:
>
> - Our sensitivity experiments with injected OCR/VLM errors (Table 7 in Appendix C.3) show a positive correlation between text clarity and global perceptual quality: when text guidance degrades, OCR $F_1$ scores and non-text SR metrics (e.g., MANIQA, CLIP-IQA) both drop, indicating that reliable text guidance benefits the entire image rather than competing with background quality.
>
> - Second, in a new analysis, our noise-map visualizations (Figure 22 in Appendix C.3) show that incorrect semantic conditioning causes the predicted diffusion noise to disperse across the whole image instead of concentrating on the specific regions that require restoration, dissipating the finite predictive capacity of the denoiser and degrading both glyph reconstruction and background textures.
>
> | Error Type | OpenOCR | GOT-OCR | MANIQA | CLIP-IQA |
> | :--- | :---: | :---: | :---: | :---: |
> | **Baseline** | 48.82 | 38.36 | 62.01 | 79.69 |
> | 30% | 38.36 (-10.46) | 28.67 (-9.69) | 45.87 (-16.14) | 63.65 (-16.04) |
> | 50% | 32.03 (-16.79) | 26.35 (-12.01) | 45.39 (-16.62) | 64.88 (-14.81) |
> | 90% | 27.52 (-21.30) | 26.35 (-12.01) | 45.61 (-16.40) | 66.00 (-13.59) |
> | Swap | 39.88 (-8.94) | 33.12 (-5.24) | 45.81 (-16.20) | 66.00 (-13.69) |
> | Drop | 41.85 (-6.97) | 32.03 (-6.33) | 44.82 (-17.19) | 65.30 (-14.39) |
>
> - Together, these results indicate that accurate, high-fidelity text guidance not only improves OCR performance but also leads to better SR quality metrics, constrains the solution space, reduces global ambiguity, and ultimately enhances both text and background restoration rather than inducing a zero-sum trade-off.
>
> ---
>
> # [Response to DpJe. Weakness 2 / Question 2]
>
> To address the Reviewer's concern, in the revised version, we have additionally conducted stress tests on GLYPH-SR under curved, dense, and multilingual text scenarios.
> Specifically, we have made the following corrections:
>
> - For irregular and curved text, we have reported results on SCUT-CTW1500 and CUTE80, which are standard stress tests for curved and perspective-distorted text, and additionally provided dedicated curved-text stress-test visualizations in Figure 19.
>
> - To further validate robustness in dense, in-the-wild settings, we have added qualitative results on highly cluttered street scenes in Figure 20.
>
> - Also we have tested for multilingual street scenes and the results have been summarized in Figure 8. It shows that our framework leverages a multilingual VLM guider (Gemini 2.5 Flash) and demonstrates consistent gains on Chinese street signs (LSVT) and Devanagari/Hindi signs (IndicSTR12-real) with representative multilingual restorations, showing that the approach can reliably restore diverse scripts in dense real-world environments.

---

> ### Author Response · Authors · 2025-11-24
>
> # [Response to DpJe. Weakness 3 / Question 1]
>
> To address the Reviewer's concern, in the revised version, we have studied the sensitivity to OCR detector quality and robustness to OCR detection errors.
> Specifically, we have made the following corrections:
>
> - A quantitative sensitivity study has been conducted in Table 7 of Appendix C.3, where we systematically inject different types and levels of OCR detection errors and measure their impact on OCR $F_{1}$ scores and perceptual metrics. The results show that all error modes degrade both readability and image quality (e.g., corrupting 50\% of characters reduces OpenOCR $F_{1}$ by 16.79 pp and MANIQA by 16.62 points), but that the model is noticeably more robust to errors that are mixed within the text or simply dropped (Swap/Drop) than to outright random false detections; importantly, even under severe noise, performance degrades but does not collapse, indicating a graceful failure behavior rather than catastrophic breakdown.
>
> - To address the concerns regarding severe degradation and VLM failures, we have conducted additional experiments in Figure 21 using real-world images with inherently noisy OCR results to demonstrate that our framework avoids catastrophic failure and that text recovery remains feasible given partial cues, even in cases where the upstream OCR detector fails to correctly detect the text. Please note from Figure 21 that, although the restoration quality can inevitably be degraded under some adverse conditions, our framework highly resilient and robust to catastrophic failure. Also, even in real-world cases of extreme low resolution where the initial guidance is inherently noisy or contains recognition errors in OCR, our framework still demonstrates an outstanding capability of recovering text shapes and details to a reasonable extent. This confirms that although severe degradation adversely affects the performance, our approach ensure sufficient resilience and strong robustness by generating meaningful restorations in practical real-world scenarios.
>
> - On top of this, to more directly explain the underlying failure modes, we have expanded the analysis in Figure 22 of Appendix C.3, which visualizes how OCR errors propagate through the denoising trajectory: when the OCR guidance is incorrect, the semantic condition conflicts with the visual evidence and the predicted noise becomes spatially dispersed rather than concentrated on the regions that require restoration, leading to degradation in both text fidelity and overall image quality. This additional ablation study links specific OCR error patterns to concrete degradation behaviors in the diffusion process, providing a more fine-grained and intuitive understanding of OCR error sensitivity beyond the aggregate numbers in Table 7.
>
> ---
>
> # [Response to DpJe. Question 3]
>
> Thank you very much for the comment.
>
> - In our framework, the ping-pong schedule is chosen empirically. Specifically, we determined the binary ping–pong schedule by comparing it against mixing strategies that blend image and text guidance. This can be shown from the results in the ablation study in Appendix C.2 (Table 4): all static mixing settings induce a sub-optimal compromise between text legibility and perceptual quality, whereas the ping–pong schedule temporally alternating between image-centric and text-centric phases consistently yields the highest OCR $F_{1}$ scores while maintaining competitive SR/IQA scores.
>
> - Although the ping-pong schedule can be learned, it would require injecting symbolic/semantic losses (e.g., OCR losses) directly into diffusion training, which tends to conflict with the pixel-level reconstruction objective and destabilize optimization. To address this issue, in our framework, we decided to keep the ping–pong schedule fixed so as not to harm the pretrained backbone as our primary design goal is to preserve the strong generative prior of this backbone while only learning to recognize and correct cases where scene text is degraded.

---

> > ### Comment · Reviewer_DpJe · 2025-11-24
> >
> > Thanks for the authors' reply. All my concerns have been resolved. Happy to raise my score.

---

> > > ### Author Response · Authors · 2025-11-24
> > >
> > > Thank you for your kind words and for taking the time to reconsider your evaluation. We're glad our responses helped clarify the concerns, and we truly appreciate your updated assessment.

---

### Official Review · Reviewer_BSKA · 2025-10-25

**Soundness:** 3
**Presentation:** 3
**Contribution:** 3
**Rating:** 6
**Confidence:** 4

**Summary:**

This paper proposed a framework GLYPH-SR, which utilizes textual information mining from the LQ to guide the denoise process through the designed TS-ControlNet. This paper also introduced ping-pong scheduler to control the condition injection strength along the denoising trajectory for better balance between visual and textual restoration. Extensive experiments demonstrate the competitive performance of the proposed framework.

**Strengths:**

Strengthens:

1. Proposed the GLYPH-SR framework with TS-ControlNet, allowing fine-grained control over both glyph-level details and scene-level realism. Furthermore, a ping-pong scheduler is introduced to dynamically balance visual fidelity and text legibility during the denoise process.
2. Constructed a factorized synthetic corpus separating text degradation from global image degradation, enabling controlled finetunning and clear ablation analysis.
3. Analyzed the trade-off between SR metrics and OCR metrics, which is essential for the evaluation of the proposed framework and similar methods.

**Weaknesses:**

Weaknesses:

1. The novelty could be further improved.
    - The text branch of the proposed TS-ControlNet continues to adopt the plain ControlNet structure, without any specific modifications for its text-focused role. Introducing task-oriented designs could potentially further improve its performance.
    - Although the paper considers the trade-off between SR and OCR metrics, it does not introduce a unified metric to evaluate both scene reconstruction and text restoration quality.

2. The experiments are insufficient. The paper lacks comprehensive evaluations on related real-world image super-resolution task.

3. Some minor writing issues. e.g. "paragraphStep" in line 273

**Questions:**

1. Does the proposed framework only support English text? How does it perform on non-Latin characters?

---

> ### Author Response · Authors · 2025-11-24
>
> # [Response to BSKA. Weakness 1-1]
>
> Thank you very much for the important comment. Through our primary experimental validation, we found that the plain ControlNet structure would rather outperform its task-specific modification. In the revised version, we have included additional results to confirm this. More elaborations and detailed discussions are provided in the following.
>
> - As illustrated in Figure 24 of the revised paper, the primary challenge in our framework is to ensure robust noise prediction, specifically focusing on resilience against incorrect text guidance and inaccurate localization in severely degraded images. That is, our critical design objective is to guide the restoration process without compromising the backbone’s performance.
>
> - To address this, we initially considered incorporating task-specific modifications into the text branch. However, it was found that introducing complex or aggressive modules tends to interfere with the pre-trained backbone, leading to performance degradation of its inherent generative capabilities. On the other hand, we also found that the plain ControlNet structure effectively isolates degraded text regions without interfering with the backbone's pre-trained features.
> Thus, in our framework, we maintained the simple design for stability.
> To confirm these, in the revised version, we have conducted additional experiments and included the results in Figure 24.
>
> - Based on the experiments and empirical demonstration, we have an important insight that a promising  task-specific modification will be to incorporate soft refinements—specifically leveraging the attention maps from the VLM's OCR module. We plan to implement this on our framework in future works, with the aim of injecting this guidance signal without compromising the backbone's pre-training performance.
>
>
> ---
>
> # [Response to BSKA. Weakness 1-2]
>
> We agree that it is necessary and useful to develop a novel unified metric evaluating both scene reconstruction and text restoration quality.
> However, such research inevitably requires to perform thorough experimental validations on the efficacy of the developed metric and also needs exhaustive comparative studies with other existing metrics. Due to this challenge, developing the unified metric remains largely unexplored in the literature (even in the most relevant works such as in [1]-[5]), despite its importance.
> For these reasons, we believe that it is beyond the scope of our current work. It should be an important next research step from our current work and deserved to be investigated further in future works.
>
> [1] Zhang et al. (2024), "Diffusion-based Blind Text Image Super-Resolution"
>
> [2] Min et al. (2025), "Text-Aware Image Restoration with Diffusion Models"
>
> [3] Xiaoming et al. (2024), "Enhanced Generative Structure Prior for Chinese Text Image Super-Resolution"
>
> [4] Chen et al. (2024), "Image Super-Resolution with Text Prompt Diffusion"
>
> [5] Li et al. (2023), "Learning Generative Structure Prior for Blind Text Image Super-resolution"
>
>
> ---
>
> # [Response to BSKA. Weakness 2]
>
> Thank you very much for the comment. We would like to discuss the issues in the following.
>
> - **Regarding the experiments:** In the revised version, we have included more results by conducting additional experiments. Specifically, additional methods [1]-[5] have been included in the experiments as well as in the related works section. In the revised paper attached, we have provided the experimental results of the methods [1]-[5] in Tables 1, 2, and 3.
>
> [1] Zhang et al. (2024), "Diffusion-based Blind Text Image Super-Resolution"
>
> [2] Min et al. (2025), "Text-Aware Image Restoration with Diffusion Models"
>
> [3] Xiaoming et al. (2024), "Enhanced Generative Structure Prior for Chinese Text Image Super-Resolution"
>
> [4] Li et al. (2023), "Learning Generative Structure Prior for Blind Text Image Super-resolution"
>
> [5] Liang et al. (2021), "SwinIR: Image Restoration Using Swin Transformer"
>
> - **Regarding the evaluation under the real-world scenario:** Furthermore, to evaluate the performance under the real-world scenario, in the revised version, we have included additional experimental results on the Real-Text dataset [6] that only contains the real degradations without manual downsampling (i.e., artificial degradations). Additional results on this real-world dataset have been presented in Figure 15 of Appendix C.1 of the revised paper. Furthermore, to facilitate a clear understanding of the real-world scene-text restoration task, we have provided extensive additional visual comparisons in Figure 4, 8, 15, 17, 19 and Figure 20 of the revised paper.
>
> [6] Jaewon Min, et al. "Text-Aware Image Restoration with Diffusion Models."

---

> ### Author Response · Authors · 2025-11-24
>
> # [Response to BSKA. Weakness 3]
>
> We thank the reviewer for pointing this out and for their careful reading. We apologize for the typographical errors, including the one noted on L273. In the revised version, suggested fixes have been addressed accordingly. Also, we have performed a thorough proofread of the entire manuscript to correct all identified typos, grammatical errors, and presentation issues to improve the paper's clarity.
>
> ---
>
> # [Response to BSKA. Question 1]
>
> The proposed GLYPH-SR is not confined only to English. It can be used to support non-Latin characters. Detailed discussions are provided in the following.
>
> - By its design mechanism, our framework separates semantic guidance from structural restoration. This makes the pipeline language-agnostic. Also, simply swapping the upstream guider enables multilingual support without retraining the core backbone as long as the backbone and OCR module support the target language.
>
> - To demonstrate this empirically, in the revised version, we have conducted additional experiment by utilizing a multilingual VLM (Gemini 2.5 Flash) to test the model on Chinese (LSVT) and Hindi (IndicSTR12) datasets. The results have been included in Figure 8.
> As shown in Figure 8, the proposed GLYPH-SR successfully recovers the complex stroke patterns of these non-Latin scripts, confirming that our framework generalizes effectively beyond standard Latin characters.

---

### Official Review · Reviewer_Yv2C · 2025-10-30

**Soundness:** 3
**Presentation:** 3
**Contribution:** 3
**Rating:** 6
**Confidence:** 4

**Summary:**

This paper introduces GLYPH-SR, a novel vision-language model (VLM)-guided latent diffusion framework designed to address the dual-objective problem in image super-resolution (SR): achieving high perceptual quality and high-fidelity scene-text recovery. The core of the method is a dual-branch Text-SR Fusion ControlNet (TS-ControlNet) that integrates scene-level captions with OCR-derived text strings and their spatial positions. A key innovation is the "ping-pong" scheduler, which dynamically alternates between text-centric and image-centric guidance during the diffusion denoising process. The model is trained on a carefully constructed synthetic corpus that factorizes glyph quality and global image quality perturbations. Extensive evaluations on SVT, SCUT-CTW1500, and CUTE80 benchmarks at up to 8× scaling demonstrate that GLYPH-SR achieves significant improvements in OCR F1 scores (up to +15.18 percentage points) while maintaining competitive performance on perceptual metrics (MANIQA, CLIP-IQA, MUSIQ) against a strong suite of diffusion and GAN-based baselines.

**Strengths:**

1. The paper compellingly argues that text legibility is a critical yet overlooked aspect of SR in practical applications. It provides a clear analysis of the systemic biases (metric and objective) in prior work that lead to text hallucination or conservative restoration, effectively framing the need for a dual-objective approach.

2. The proposed TS-ControlNet architecture and the binary ping-pong scheduler are elegant and effective solutions for fusing semantic text cues with global image priors without disrupting the pre-trained diffusion backbone. The design allows for targeted text restoration through fine-tuning a relatively small number of parameters.

3. The construction of a four-partition synthetic dataset is a significant methodological contribution. It enables the precise disentanglement of text restoration from general SR, providing a clean signal for training the text-specific components.

4. The paper provides an extensive empirical evaluation across multiple datasets, scale factors, and a wide range of state-of-the-art baselines. The dual-axis evaluation protocol, reporting both OCR metrics and perceptual quality metrics, is thorough and appropriate for the claimed contributions.

5. The ablation studies on guidance components, the scheduler policy, and the sensitivity analysis to upstream OCR/VLM errors are systematic and provide valuable insights into the model's behavior, strengths, and limitations.

6. The qualitative results (Figures 1, 4, 5, 12, 13) are highly effective. They clearly demonstrate GLYPH-SR's superior ability to reconstruct legible, accurate text in challenging scenarios (e.g., 8× scaling) where other methods fail, providing strong visual support for the quantitative claims.

**Weaknesses:**

1. The related work, experiment  section (Section 2/4) and lacks a thorough discussion of several recent and highly relevant works that also leverage VLMs, text prompts, or diffusion models for text-aware image restoration. Notable omissions include, but are not limited to:

    a) Zhang et al. (2024), "Diffusion-based Blind Text Image Super-Resolution"

    b) Chen et al. (2024), "Image Super-Resolution with Text Prompt Diffusion" / "Universal Image Restoration with Text Prompt Diffusion"

    c) Zhang et al. (2024), "ConsisSR: Delving Deep into Consistency in Diffusion-based Image Super-Resolution"

    d) Bogolin (2025), "Text-Aware Image Restoration with Diffusion Models"

    e) Xiaoming et al. (2024), "Enhanced Generative Structure Prior for Chinese Text Image Super-Resolution"

    This gap weakens the contextualization of the paper's novelty and leaves the reader uncertain about how GLYPH-SR specifically advances the field beyond these concurrent efforts. A more comprehensive survey and a clearer delineation of contributions are needed.

2. While Figure 14 is provided, the discussion of failure cases is somewhat brief. A deeper analysis is warranted, particularly regarding: (a) the root cause of text hallucination in non-text regions (e.g., is it due to over-reliance on text guidance or errors in the initial OCR?); (b) the model's tendency to enhance only the most salient text instances; and (c) a critical failure mode not explicitly discussed: what happens when the upstream VLM/OCR fails to detect or correctly recognize severely degraded text in the LR input? This scenario is highly probable in real-world applications and likely breaks the method's core premise.

3. As shown in Table 6, GLYPH-SR's computational footprint (13B+ parameters, ~43GB VRAM, ~38s/inference) is substantial, limiting its practical deployability compared to faster baselines. The discussion on potential efficiency improvements (Section C.4) is preliminary and speculative. A more concrete analysis or preliminary results from, for example, a distilled VLM, would strengthen the paper's practical impact.

4. Minor Typos and Presentation: L273. The authors should carefully check the content.

**Questions:**

1. The sensitivity analysis in Table 5 uses simulated OCR errors. How does GLYPH-SR perform on real-world low-quality images where the initial OCR (providing S_TXT) is inherently noisy or incomplete? Can you provide results on a wild dataset with poor ground-truth OCR to demonstrate robustness?

2. Could you provide more details on the tuning of critical hyperparameters like the control scale s_CTRL and the CFG scale ω? Were they empirically tuned, and what are the observed trade-offs between text fidelity and image quality at different values? Are there failure modes associated with extreme values?

3. Have you conducted any experiments with a smaller or quantized VLM (e.g., a distilled version of LLaVA-NeXT) to reduce computational cost? If so, what was the corresponding drop in OCR F1 and perceptual scores? This would greatly inform practical applications.

4. The paper rightly notes the misalignment of traditional metrics (Fig. 7). To further substantiate the perceptual claims, have you considered or conducted a user study to quantitatively assess human preference between GLYPH-SR and key baselines regarding both overall image quality and text readability?

5. The evaluation focuses on standard Latin scripts. What are the prospects or any preliminary results for GLYPH-SR on multilingual text, complex scripts (e.g., Chinese, Arabic), or handwritten text? Does the current design have inherent limitations for such scenarios?

6. Based on the analysis of Figure 14, what specific architectural or training modifications (e.g., incorporating a text-region segmentation mask, adding a localization loss, or using a more robust text detector) do you envision could mitigate the issues of hallucination in non-text regions and incomplete enhancement of multiple text instances?

---

> ### Author Response · Authors · 2025-11-24
>
> # [Response to Yv2C. Weakness 1]
>
> In the revised version, we have improved our work based on the Reviewer's important comment and valuable suggestion. Detailed discussions are provided in the following.
>
> - In the revised version, we have improved the related works section. Specifically, in Section 2 of the revised paper, we have included the recent and highly relevant works a)-e), [1], and [2], and provided thorough and comprehensive discussions on these works and our work. Roughly speaking, the works a)-e), [1], and [2] can be categorized into three groups based on their intrinsic limitations: (i) Cropped-Text SR (a, d, [1]) limited to cropped inputs, (ii) General-Purpose SR (b-1,b-2, c, [2]) lacking explicit text-aware architecture, and (iii) General Text-Aware LR Restoration (e) targeting broad restoration rather than the information-scarce SR task.
>
> [1] Li et al. (2023), "Learning Generative Structure Prior for Blind Text Image Super-resolution"
>
> [2] Liang et al. (2021), "SwinIR: Image Restoration Using Swin Transformer"
>
> - It is important to note that our work is quite different from the works a)-e), [1], and [2] in several aspects. The differences are concisely summarized in the following table:
>
> $$
> \\begin{array}{l|cccccccc|c}
> \\hline
> \\textbf{Method} & \\text{a)} & \\text{b-1)} & \\text{b-2)} & \\text{c)} & \\text{d)} & \\text{e)} & \\text{[1]} & \\text{[2]} & \\textbf{Ours} \\\\
> \\hline
> \\text{Text-aware SR} & \\textsf{O} & \\textsf{X} & \\textsf{X} & \\textsf{X} & \\textsf{O} & \\textsf{O} & \\textsf{O} & \\textsf{X} & \\textbf{\\textsf{O}} \\\\
> \\text{Gen-Purpose SR} & \\textsf{X} & \\textsf{O} & \\textsf{O} & \\textsf{O} & \\textsf{X} & \\textsf{O} & \\textsf{X} & \\textsf{O} & \\textbf{\\textsf{O}} \\\\
> \\text{LR Robustness} & \\textsf{O} & \\textsf{O} & \\textsf{X} & \\textsf{O} & \\textsf{O} & \\textsf{X} & \\textsf{O} & \\textsf{O} & \\textbf{\\textsf{O}} \\\\
> \\hline
> \\end{array}
> $$
>
> - Also, in the revised version, we have included more results by conducting additional experiments with the most relevant  works in a), d), e), [1], and [2]. We have provided the experimental results with these competing methods in Tables 1, 2, and 3.
>
> ---
>
> # [Response to Yv2C. Weakness 2]
>
> Thank you very much for the valuable suggestion. In the revised version, we have improved the analysis, and provided thorough and more detailed discussions of failure cases in the following aspects:
>
> - **(a) Root cause of hallucination in non-text regions:** To clarify why wrong OCR leads to hallucinations, in the revised paper, we have further analyzed the denoising trajectory with controlled OCR perturbations and visualized the predicted noise maps. The results have been included in Appendix C.3 (Figure 22). It can be seen that when the OCR guidance is incorrect, the semantic prior conflicts with the visual prior and the noise prediction becomes spatially dispersed instead of focusing on the true text regions,  exhibiting how the erroneous guidance corrupts nearby background content.
>
> - **(b) Tendency to enhance only the most salient text instances:**  In very low-quality images, the upstream OCR often produces noisy or incomplete tokens. In this case, the backbone model should implicitly decide where this guidance has to be attached. In practice, it anchors the text prior to the best-visible word: cross-attention assigns higher weight to the most legible or high-contrast region, and the limited denoising budget is spent primarily on that glyph (i.e., text shape information), while more degraded or faint text receives weaker updates.
>
> - **(c) Behavior when the upstream VLM/OCR fails:** To address the concerns regarding severe degradation and real-world VLM failures, in the revised version, we also have conducted additional experiments using real-world low-quality images with inherently noisy and incomplete VLM OCR results. It has been demonstrated in Figure 21.
> Please note from Figure 21 that, although the restoration quality can inevitably be degraded under some adverse conditions, our framework highly resilient and robust to catastrophic failure. Also, even in real-world cases of extreme low resolution where the initial guidance is inherently noisy or contains recognition errors in OCR, our framework still demonstrates an outstanding capability of recovering text shapes and details to a reasonable extent. This confirms that although severe degradation adversely affects the performance, our approach ensure sufficient resilience and strong robustness by generating meaningful restorations in practical real-world scenarios.

---

> ### Author Response · Authors · 2025-11-24
>
> # [Response to Yv2C. Weakness 3 / Question 3]
>
> Thank you very much for the important comment and valuable suggestion. We absolutely agree with this point. We would like to discuss the issues in the following.
>
> - To address the comment, in the revised version, we have conducted additional  experiments on ablation studies with compact, lighter VLMs (e.g., involving 3B, 4B parameters) by varying OCR capabilities. The results have been included in Table 9 of the revised paper. We have also provided a concrete analysis on the results. Our analysis reveals that replacing the guider with a lighter module having significantly lower OCR capabilities leads to degradation in final restoration quality, suggesting that a high-capacity (yet heavy) VLM may still be crucial for effectively coping with severe degradations, albeit at the cost of high computational burden.
>
> $$
> \\begin{array}{l|ccc|ccc}
> \\hline
> \\textbf{Model} & \\text{OpenOCR} & \\text{GOT-OCR} & \\text{LLaVA} & \\text{MANIQA} & \\text{CLIP-IQA} & \\text{MUSIQ} \\\\
> \\hline
> \\textbf{Ours (8B)} & \\textbf{73.09} & \\textbf{55.62} & \\textbf{85.01} & \\textbf{49.77} & \\textbf{65.93} & \\textbf{69.96} \\\\
> \\text{YannQi/R-4B} (4B) & 72.77 & 55.21 & 84.75 & 40.78 & 57.20 & 65.57 \\\\
> \\text{google/paligemma-3b-mix-448 (3B)} & 70.53 & 51.45 & 84.52 & 42.11 & 59.61 & 65.83 \\\\
> \\hline
> \\end{array}
> $$
>
> - Through the analysis and empirical demonstration, we also found an important consequence that VLM models of modest sizes can achieve acceptable performance while striking a practical and balanced trade-off between accuracy and efficiency. Thus these models will be useful for resource-constrained environments. The results of Table 9 summarizes the accuracy–efficiency trade-off trends and demonstrates that GLYPH-SR can be adapted to different computation/deployment budgets without critically degrading reliability.
>
> ---
>
> # [Response to Yv2C. Weakness 4]
>
>
> We thank Reviewer for pointing this out and for their careful reading. We apologize for the typographical errors, including the one noted on L273. In the revised version, suggested fixes have been addressed accordingly. Also, we have performed a thorough proofread of the entire manuscript to correct all identified typos, grammatical errors, and presentation issues to improve the paper's clarity.

---

> ### Author Response · Authors · 2025-11-24
>
> # [Response to Yv2C. Question 1]
>
> To address the Reviewer's concern, in the revised version, we have studied the sensitivity to OCR detector quality and robustness to OCR detection errors.
> Also, we have included how GLYPH-SR performs on real-world low-quality images.
> Specifically, we have improved the paper in the following aspects:
>
> - A quantitative sensitivity study has been conducted in Table 7 of Appendix C.3, where we systematically inject different types and levels of OCR detection errors and measure their impact on OCR $F_{1}$ scores and perceptual metrics. The results show that all error modes degrade both readability and image quality (e.g., corrupting 50\% of characters reduces OpenOCR $F_{1}$ by 16.79 pp and MANIQA by 16.62 points), but that the model is noticeably more robust to errors that are mixed within the text or simply dropped (Swap/Drop) than to outright random false detections; importantly, even under severe noise, performance degrades but does not collapse, indicating a graceful failure behavior rather than catastrophic breakdown.
>
> - To address the concerns regarding severe degradation and VLM failures, we have conducted additional experiments in Figure 21 using real-world images with inherently noisy OCR results to demonstrate that our framework avoids catastrophic failure and that text recovery remains feasible given partial cues, even in cases where the upstream OCR detector fails to correctly detect the text.
>
> - On top of this, to more directly explain the underlying failure modes, we have expanded the analysis in Figure 22 of Appendix C.3, which visualizes how OCR errors propagate through the denoising trajectory: when the OCR guidance is incorrect, the semantic condition conflicts with the visual evidence and the predicted noise becomes spatially dispersed rather than concentrated on the regions that require restoration, leading to degradation in both text fidelity and overall image quality. This additional ablation study links specific OCR error patterns to concrete degradation behaviors in the diffusion process, providing a more fine-grained and intuitive understanding of OCR error sensitivity beyond the aggregate numbers in Table 7.
>
> - Also, to evaluate the performance under the real-world scenario, in the revised version, we have included additional experimental results on the Real-Text dataset [1] that only contains the real degradations without manual downsampling (i.e., artificial degradations). The results on this real-world dataset have been presented in Figure 15 of Appendix C.1 of the revised paper. Furthermore, to facilitate a clear understanding of the real-world scene-text restoration task, we have provided extensive additional visual comparisons in Figures 4, 8, 15, 17, 19, and 20 of the revised paper.
>
> [1] Jaewon Min, et al. "Text-Aware Image Restoration with Diffusion Models."

---

> ### Author Response · Authors · 2025-11-24
>
> # [Response to Yv2C. Question 2]
>
> Thank you very much for the important comment. We would like to discuss the issues in the following aspects:
>
> - **Regarding the hyperparameter selection:** In our framework, both the CFG scale $w$ and the ControlNet scale $s_{\text{CTRL}}$ were tuned empirically to balance text fidelity and perceptual quality.
>
> - **Regarding the trade-offs:** Through the experimental results, we observe that increasing $w$ generally improves SR/IQA metrics but, beyond a moderate range, starts to degrade OCR $F_1$ scores, while large $s_{\text{CTRL}}$ values lead to over-conditioning and unstable images (including clear failure modes at extreme settings). Based on these trends, we adopt $w=7.5$ and $s_{\text{CTRL}}=1.0$ as our default operating point, which provides a good trade-off without triggering catastrophic artifacts. In the revised version, we have included the full ablation study on the trade-offs and reported their quantitative results across diverse settings in Appendix C.2 (Tables 5 and 6).
>
> - **Regarding the failure modes:** Regarding failure modes, we observe catastrophic collapse in restoration quality when $s_{CTRL}$ exceeds 2.0, and significant degradation in text fidelity at extreme CFG scales ($w \ge 12$).
>
> $$
> \\begin{array}{c|ccc|ccc}
> \\hline
> \\textbf{CFG } (w) & \\text{OpenOCR} & \\text{GOT} & \\text{LLaVA} & \\text{MANIQA} & \\text{CLIP} & \\text{MUSIQ} \\\\
> \\hline
> 2 & \\textbf{75.54}&\\textbf{57.62}&87.53&41.26&55.96&65.23\\\\
> 4&74.63&57.22&\\textbf{88.50}&43.42&60.01&67.30\\\\
> \\textbf{7.5}^*&\\underline{73.09}&\\underline{55.62}&\\underline{85.01}&\\underline{49.77}&\\underline{65.93}& \\underline{69.96}\\\\
> 12&71.50&53.56&84.23&49.52&69.87&70.51\\\\
> 20&65.45&45.18&79.06&\\textbf{50.60}&\\textbf{74.30}&\\textbf{71.03}\\\\
> \\hline
> \\end{array}
> $$
>
> $$
> \\begin{array}{c|ccc|ccc}
> \\hline
> \\textbf{Scale} (s_{CTRL})&\\text{OpenOCR}&\\text{GOT}&\\text{LLaVA}&\\text{MANIQA}&\\text{CLIP}&\\text{MUSIQ}\\\\
> \\hline
> \\textbf{1.0}^*&\\underline{\\textbf{73.09}}&\\underline{\\textbf{55.62}}&\\underline{\\textbf{85.01}}& \\underline{\\textbf{49.77}}&\\underline{65.93}&\\underline{\\textbf{69.96}} \\\\
> 2.0&69.21&56.02&81.84&26.36&\\textbf{66.95}&49.92\\\\
> 3.0&4.56&6.04&25.17&37.17&45.28&49.21\\\\
> 10.0&0.00&0.00&0.00&35.22&25.26&27.18\\\\
> \\hline
> \\end{array}
> $$
>
> ---
>
> # [Response to Yv2C. Question 4]
>
> Thank you very much for the constructive comment. We have not conducted such a user study yet for the following reason:
>
> - To our knowledge, a useful and unified metric evaluating both scene reconstruction and text restoration quality has not been developed yet in the literature including the most relevant works a)-e), [1], and [2]. We believe that it would be more convincing and effective to conduct the user study after developing the meaningful unified metric.
>
> - Even if such a metric is available, it may be practically difficult to complete the user study during this rebuttal period due to the strict time limit. In general, it is time-consuming, involved, and costly.
>
> For the reasons above, we believe that the user study is beyond the scope of our current work. It should be an important next research step from our current work and deserved to be investigated further in future works.
>
>
> ---
>
> # [Response to Yv2C. Question 5]
>
> Thank you very much for the comment. GLYPH-SR can be used to support non-Latin characters. Detailed discussions are provided in the following.
>
> - By its design mechanism, our framework separates semantic guidance from structural restoration. This makes the pipeline language-agnostic. Also, simply swapping the upstream guider enables multilingual support without retraining the core backbone as long as the backbone and OCR module support the target language.
>
> - To demonstrate this empirically, in the revised version, we have conducted additional experiment by utilizing a multilingual VLM (Gemini 2.5 Flash) to test the model on Chinese (LSVT) and Hindi (IndicSTR12) datasets. The results have been included in Figure 8.
> As shown in Figure 8, the proposed GLYPH-SR successfully recovers the complex stroke patterns of these non-Latin scripts, confirming that our framework generalizes effectively beyond standard Latin characters.
>
> - Furthermore, through additional experimental validation, we have empirically demonstrate that our framework operates effectively even on handwritten text. The results have been included in Figure 17, confirming the reliability of GLYPH-SR regardless of writing style.
> More findings and discussions have been included in Section 4 and Appendix C of the revised paper.

---

> ### Author Response · Authors · 2025-11-24
>
> # [Response to Yv2C. Question 6]
>
> Thank you very much for the important comment. We would like to discuss the issues in the following aspects:
>
> - **Incorporating a text-region segmentation mask:** We found that this approach unfortunately tends to interfere with the pre-trained backbone, leading to performance degradation of its inherent generative capabilities. On the other hand, the plain ControlNet structure effectively isolates degraded text regions without interfering with the backbone’s pre-trained features, meaning that it would rather outperform its task-specific modification. Thus, in our framework, we maintained the simple design for stability. To confirm these, in the revised version, we have conducted additional experiments and included the results in Figure 24.
>
> - **Adding a localization loss:** In our framework, the localization loss is already included in the OCR module, but not in the diffusion model. Although one might try to incorporate an explicit localization loss directly into the diffusion model, its efficacy and merits/demerits have not been fully veiled (even in terms of experimental validation), to our knowledge. Therefore, at this time, we cannot envision the impact of adding the localization loss to the diffusion model, on our framework.
> Such a research topic actually remains largely unexplored in the literature. It should be an important next research step from our current work and deserved to be investigated further in future works.
>
> - **Using a more robust text detector:** In the revised version, this approach has been empirically demonstrated through diverse experimental validations. The quantitative results have been reported in Tables 7 and 9, and the visualization results have been shown in Figure 22.
> Based on these results along with our analysis, it is confirmed that both OCR performance and image quality metrics improve as the text detector becomes more robust.
>
> - Overall, based on the discussions above, we envision (or conjecture) that the robust text detection approach is most effective and critical to address the issues. In practice, employing a more powerful OCR module to provide clearer and more precise guidance will not only improve these two metrics, but also effectively mitigate specific issues, such as hallucinations in non-text regions and the incomplete enhancement of multiple text instances.

---

> > ### Comment · Reviewer_Yv2C · 2025-11-28
> >
> > Thank you for the authors’ response. Overall, I feel that the authors are still primarily reiterating the conclusions presented in the paper and emphasizing the current limitations of the model, but have not offered clear directions on how to address them. For example, how to mitigate potential errors in text detection or how to handle the inherent trade-offs. While describing the phenomenon is valuable, proposing actionable solutions is what matters most at this stage.
> >
> > While I acknowledge some of the contributions made in this paper, I do not view them as a definitive breakthrough. Therefore, I am inclined to maintain my current score.

---

> > > ### Author Response · Authors · 2025-12-03
> > >
> > > We sincerely thank Reviewer for the valuable feedback and the comment. To address the concern, in the revised version, we have improved the conclusions from our work by providing further research directions to address the issues (rather than simply mentioning the limitations), including how to mitigate potential errors in text detection, how to handle the inherent trade-offs, and suggestion of actionable solutions, within comprehensive and empirical scopes.
> > >
> > > Specifically, in the revised version, we have suggested two main actionable solutions and related research directions to overcome the limitations, as detailed below.
> > >
> > > - **Integration of Dual-Stage Restoration [1]:** This strategy is to mitigate hallucinations and detection errors.
> > >
> > > - **Chain-of-Thought (CoT) Prompting:** This strategy is to enhance prompt precision and handle trade-offs by refining the reasoning between extracted OCR and captions.
> > >
> > >
> > > In the revised version, we have also tested the efficacy of the above two specific strategies by conducting a proof-of-concept experiment under an extreme setting.
> > >
> > > Specifically, we have performed the SR experiment on the CUTE80 dataset with a downscaling factor of $\times 16$. In this setting, the low-resolution images are severely degraded and arguably illegible even to the human eye, which inevitably leads to severe errors in text detection.
> > >
> > > The results have been reported in Table 10 in the revised paper (shown below), based on which we could conjecture that even in such an extreme $\times 16$ environment, our suggested solutions and research directions yield remarkable performance improvements and have the potentials to break through the limitations in the following aspects:
> > >
> > >  - **Mitigating Detection Errors (GLYPH-SR + [1]):** Incorporating the dual-stage approach from [1] resulted in a dramatic improvement in text recognition accuracy (OpenOCR score improved from 9.63 to 13.77). This demonstrates that structural guidance is a viable solution for reducing hallucinations and recovering text structure when detection fails.
> > >
> > >  - **Handling Trade-offs (GLYPH-SR + CoT):** Refining the prompt via Chain-of-Thought reasoning significantly boosted perceptual quality metrics (ManiQA $\uparrow$ 3.63, ClipIQA $\uparrow$ 5.31) and improved recognition (GOT-OCR $\uparrow$ 1.37). This confirms that better semantic understanding can guide the model to balance the trade-off between fidelity and perception more effectively.
> > >
> > > | Method | OpenOCR | GOT-OCR | ManiQA | ClipIQA |
> > > | :--- | :---: | :---: | :---: | :---: |
> > > | LR | 12.41 | 12.41 | 14.12 | 29.11 |
> > > | GLYPH-SR (Baseline) | 9.63 | 11.72 | 38.52 | 54.68 |
> > > | GLYPH-SR + [1] | **13.77** $\color{red}{(+4.14)}$ | 12.41 $\color{red}{(+0.69)}$ | 35.75 $\color{blue}{(-2.77)}$ | 49.68 $\color{blue}{(-5.00)}$ |
> > > | GLYPH-SR + CoT | 10.33 $\color{red}{(+0.70)}$ | **13.09** $\color{red}{(+1.37)}$ | **42.15** $\color{red}{(+3.63)}$ | **59.99** $\color{red}{(+5.31)}$ |
> > >
> > > *Preliminary evaluation on CUTE80 ($\times 16$).*
> > >
> > > The above empirical validation further confirms the robust extendability and scalability of our framework (GLYPH-SR) in handling extreme degradation by effectively mitigating detection errors and balancing trade-offs.
> > >
> > > [1] "A Novel VLM-Guided Diffusion Model for Remote Sensing Image Super-Resolution." IEEE Geoscience and Remote Sensing Letters (2025).

---

### Official Review · Reviewer_yiSZ · 2025-11-01

**Soundness:** 2
**Presentation:** 2
**Contribution:** 2
**Rating:** 4
**Confidence:** 4

**Summary:**

This paper proposes GLYPH-SR, a vision–language guided diffusion framework for text image super-resolution, aiming to jointly optimize image perceptual quality and text legibility.
The core technical contributions include:
1. Bi-objective formulation and dual-axis protocol that treats SR as joint optimization of image and text fidelity.
2. Text-SR Fusion ControlNet, which fuses OCR-derived textual cues and image captions.
3. A ping-pong scheduler alternating text-centric and image-centric guidance during diffusion denoising.
4. A synthetic factorized corpus that decouples text and image degradation for targeted training.

**Strengths:**

1. The bi-objective view of SR (visual realism + text fidelity) is intuitive and important for practical use, addressing the neglected fact in most STISR works.
2. The TS-ControlNet + ping-pong scheduler combination is intuitive for the target of optimizing image perceptual quality and text legibility jointly.
3. The four-way partition synthetic corpus fits the claimed objective of joint optimization for training purpose.

**Weaknesses:**

1. Most baselines in experiments are not SR methods specialized for scene text image, except DiffTSR. In addition, methods like DiffTSR are not built for restoring a full scene text image, but for cropped image that only contains a single textline. The comparison could be unfair.

2. Despite most baselines were not built for scene text image, the proposed GLYPH-SR still can not outperform them consistently, even in terms of OCR accuracy.

3. As mentioned in Sec. C.3, the restoration performance rely heavily on the OCR result at the beginning of the procedure. Though strong VLM is applied, the OCR result could still be wrong under severe degradation, otherwise the super-resolution is unnecessary.

4. The dataset used for training and **evaluation** is not specifically built for image super-resolution. Even in the evaluation, the LR images seem to be generated by manually downsampling. The lack of real-world scenario in evaluation made it less convincing.

5. The OCR accuracy on LR/HR image is not reported in the tables, which made it harder to see the improvements.

**Questions:**

1. How was DiffTSR applied to this task? Were the full scene text images directly fed to DiffTSR, or fed after cropping?

2. Why is this paper named "GLYPH-SR" ? The Text-guidance in ControlNet only contains the recognized text and detected position. It seems that this work has nothing to do with glyph.

3. What is the OCR accuracy on the original image/downsampled(HR/LR) image?

4. Have the authors consideder applying other methods to balance the image and text guidance instead of the binary ping-pong policy? (e.g. through a dynamic learnable parameter)

---

> ### Author Response · Authors · 2025-11-24
>
> # [Response to yiSZ. Weakness 1 / Question 1]
>
> Thank you very much for the comment. We would like to discuss the issues in the following aspects:
>
> - **Regarding the baseline SR methods:** In the revised version, additional SR methods [1]-[3] specialized for scene text image have been included in the experiments as well as in the related works section. The additional experimental results of these SR methods have been reported in Tables 1, 2, and 3.
>
> [1] Min et al. (2025), "Text-Aware Image Restoration with Diffusion Models"
>
> [2] Xiaoming et al. (2024), "Enhanced Generative Structure Prior for Chinese Text Image Super-Resolution"
>
> [3] Li et al. (2023), "Learning Generative Structure Prior for Blind Text Image Super-resolution"
>
> - **Regarding the implementation of DiffTSR:** In the experiments, the full scene text images were directly fed to DiffTSR without cropping. This is to demonstrate that text-specialized models lacking global context awareness are insufficient for full-scene restoration.
> The same approach is also applied to other text-specific SR methods [1]-[3].
>
> ---
> # [Response to yiSZ. Weakness 2]
>
> Thank you very much for the important comment.
>
> - We would like to note that even among the existing baselines, there exist inherent tradeoffs: no method outperforms the others in all metrics, e.g., as can be seen from the results in Table 1 of the paper. Even in terms of OCR accuracy, such tradeoffs can still be observed:  if one baseline method performs well on a particular OCR model, it tends to perform poorly on other OCR models. Due to this fundamental tradeoff issue, the proposed GLYPH-SR could not outperform all baselines consistently.
>
> - Nevertheless, our framework excels in almost all performance metrics with remarkable performance improvements, exhibiting better tradeoffs than the baselines.
> To confirm this, we have counted the number of achieving the best or second-best performance for each method and attached the results in Table 1 of the revised version. It can be seen that the number of times the proposed GLYPH-SR achieves the best or second-best performance is largest, demonstrating the efficacy and supremacy of our approach.
>
> | Model | GLYPH-SR | DiffBIR | TAIR | Real-ESRGAN | PiSA-SR | StableSR | SwinIR | BSRGAN | InvSR | SUPIR |
> | :--- | :---: | :---: | :---: | :---: | :---: | :---: | :---: | :---: | :---: | :---: |
> | **1st Rank** | **20** | 6 | 4 | 3 | 1 | 1 | 1 | 0 | 0 | 0 |
> | **2nd Rank** | **8** | 6 | 4 | 2 | 3 | 3 | 0 | 4 | 4 | 3 |
>
> *Other models (DiffTSR, MARCONet, etc.) received 0 votes.*
>
> ---
> # [Response to yiSZ. Weakness 3]
>
> Thank you very much for the comment. We would like to discuss the issue in the following.
>
> - We agree that the OCR result could be wrong under severe degradation. To overcome adverse circumstances under severe degradation, therefore, the super-resolution will be required for OCR in practice. To examine this, in the revised version, we have conducted additional experiments on real-world low-quality images where the initial OCR guidance contains recognition errors due to severe degradation. The results have been provided in Figure 21. It demonstrates that even under such situation with compromised guidance, our framework still maintains a certain degree of resilience and robustness, revealing the practical feasibility of text super-resolution, and thereby, envisioning a synergetic complement between super-resolution and OCR.
>
> - By the way, to our knowledge, it is not fully veiled how the super-resolution affects the OCR result. It also remains largely unexplored how OCR and super-resolution complement each other and create synergy effects. These research topics are deserved to be investigated further in the literature. Our work is an effort in this research direction.

---

> ### Author Response · Authors · 2025-11-24
>
> # [Response to yiSZ. Weakness 4]
>
> Thank you very much for the comment.
>
> - **Regarding the datasets and manual downsampling:** Existing datasets for image super-resolution do not contain enough texts to validate and evaluate the efficacy of our framework. To address this issue, we adopt the widely used, text-rich OCR datasets (e.g., SVT, SCUT-CTW1500, and CUTE80) that contain enough texts to validate our approach. However, these OCR datasets cannot be directly used to validate the effectiveness and performance of our framework in image super-resolution as they do not provide low-resolution images.
> To obtain text-rich low-resolution images for performance evaluation and validation of our framework, we manually degrade the resolution of images in the adopted OCR datasets.
> Please note that this approach (i.e., manually degrading the resolution of images) is not limited to our work, but is very common and widely adopted in the literature when the low-resolution images are not available.
>
> - **Regarding the evaluation under the real-world scenario:** To evaluate the performance under the real-world scenario, in the revised version, we have included additional experimental results on the Real-Text dataset [1] that only contains the real degradations without manual downsampling (i.e., artificial degradations). The results on this real-world dataset have been presented in Figure 15 of Appendix C.1 of the revised paper. Furthermore, to facilitate a clear understanding of the real-world scene-text restoration task, we have provided extensive additional visual comparisons in Figures 4, 8, 15, 17, 19, and 20 of the revised paper.
>
> [1] Jaewon Min, et al. Text-Aware Image Restoration with Diffusion Models.
>
> ---
> # [Response to yiSZ. Weakness 5 / Question 3]
>
> We sincerely thank Reviewer for this valuable suggestion.
> In the revised version, we have reported the OCR accuracy for LR and HR images.
>
> Specifically, in the updated experimental results, we have added the OCR accuracy metrics for the LR images in Table 1 to provide a direct baseline for comparison. Furthermore, we have explicitly included comprehensive OCR accuracy results and image quality results for both LR and HR images in Tables 2 and 3, enabling a more detailed assessment of the performance gains relative to the input and the ground truth.
>
> ---
> # [Response to yiSZ. Question 2]
>
> Thank you very much for the comment. Most existing SR methods just treated the text in the image as texture without careful consideration or inspection, missing the meaningful and useful shape information of the text (i.e., glyph).
> By contrast, our framework preserves the text shape information during the image SR. For this reason, we named our framework as "GLYPH-SR", highlighting its dual capability of capturing the text shape information (even restoring the specific shapes and legibility of the text) while achieving the image SR.
>
> Nevertheless, as Reviewer commented, this name might seem immature. If you suggest more appropriate name of our framework, we will gladly accommodate it.
>
> ---
> # [Response to yiSZ. Question 4]
>
> Certainly, we also applied other methods based on directly mixing the two guidance conditions (image and text) by varying mixing ratios. The experimental results and ablation study for this approach were provided in Appendix C.2 (Table 4) in the submitted version, showing that the static mixing strategy consistently yields a poorer balance between perceptual quality and text legibility than the binary ping–pong scheduler.
>
> $$
> \\begin{array}{l|cccc}
> \\hline
> \\text{Scheduler Policy}&\\text{MANIQA} & \\text{CLIP-IQA} & \\text{MUSIQ} & \\text{OCR } F_1 \\\\
> \\hline
> \\textbf{(a) CUTE80 (LR}\\times\\textbf{ 4)} & & & & \\\\
> \\hline
> \\text{Binary ping–pong} & 49.77 & 65.93 & 69.96 & \\textcolor{red}{\\underline{\\textbf{85.01}}} \\\\
> \\text{Mixing } (\\lambda_{t}=0.1) & \\textcolor{red}{\\underline{\\textbf{49.95}}} & \\textcolor{red}{\\underline{\\textbf{70.64}}} & \\textcolor{red}{\\underline{\\textbf{70.67}}} & 81.57 \\\\
> \\text{Mixing} (\\lambda_{t}=0.3)&49.04&69.56& 69.75 & 83.18 \\\\
> \\text{Mixing} (\\lambda_{t}=0.5)&47.57&65.47& 68.95 & 84.23 \\\\
> \\text{Mixing} (\\lambda_{t}=0.7)&47.86&68.91&68.83 & 81.84 \\\\
> \\text{Mixing} (\\lambda_{t}=0.9)&48.85&69.11&69.13 & 82.65 \\\\
> \\hline
> \\textbf{(b) CUTE80 (LR } \\times \\textbf{ 8)} & & & & \\\\
> \\hline
> \\text{Binary ping–pong}&47.75&65.85 & 68.85&\\textcolor{red}{\\underline{\\textbf{73.71}}} \\\\
> \\text{Mixing} (\\lambda_{t}=0.1)&\\textcolor{red}{\\underline{\\textbf{48.89}}}&67.65&\\textcolor{red}{\\underline{\\textbf{69.56}}}&66.49 \\\\
> \\text{Mixing} (\\lambda_{t}=0.3)&47.44&\\textcolor{red}{\\underline{\\textbf{68.31}}}&68.86&69.87\\\\
> \\text{Mixing} (\\lambda_{t}=0.5)&46.57&64.07&67.35&73.40 \\\\
> \\text{Mixing} (\\lambda_{t}=0.7)&45.80&67.98&67.19&66.84 \\\\
> \\text{Mixing} (\\lambda_{t}=0.9)&45.58&67.66&67.18&68.88 \\\\
> \\hline
> \\end{array}
> $$

---

### Note · Authors · 2026-01-04

I have read and agree with the venue's withdrawal policy on behalf of myself and my co-authors.